# Extracellular matrix educates an immunoregulatory tumor macrophage phenotype found in ovarian cancer metastasis

E. H. Puttock [1,5], E. J. Tyler [1,5], M. Manni[2], E. Maniati [1], C. Butterworth [1], M. Burger Ramos[1], E. Peerani [1], P. Hirani[1], V. Gauthier [1], Y. Liu[1], G. Maniscalco[1], V. Rajeeve[1], P. Cutillas [1], C. Trevisan[3], M. Pozzobon[3], M. Lockley [1], J. Rastrick[4], H. Läubli [2], A. White[4] & O. M. T. Pearce [1] ✉

Recent studies have shown that the tumor extracellular matrix (ECM) associates with immunosuppression, and that targeting the ECM can improve immune infiltration and responsiveness to immunotherapy. A question that remains unresolved is whether the ECM directly educates the immune phenotypes seen in tumors. Here, we identify a tumor-associated macrophage (TAM) population associated with poor prognosis, interruption of the cancer immunity cycle, and tumor ECM composition. To investigate whether the ECM was capable of generating this TAM phenotype, we developed a decellularized tissue model that retains the native ECM architecture and composition. Macrophages cultured on decellularized ovarian metastasis shared transcriptional profiles with the TAMs found in human tissue. ECM-educated macrophages have a tissue-remodeling and immunoregulatory phenotype, inducing altered T cell marker expression and proliferation. We conclude that the tumor ECM directly educates this macrophage population found in cancer tissues. Therefore, current and emerging cancer therapies that target the tumor ECM may be tailored to improve macrophage phenotype and their downstream regulation of immunity.

The remodeling of the extracellular matrix (ECM) that accompanies invasive carcinomas is associated with the establishment of an immunosuppressive environment and poor clinical response to immunotherapy[1,2]. Preclinical models where tumor ECM is targeted either through non-specific[3,4] or specific[5] inhibition have been shown to support anti-tumor immunity and improve response to immunotherapy. For example, in a preclinical model of colorectal carcinoma, broad-spectrum inhibitors of ECM deposition, such as the TGF-β inhibitor galunisertib, reduced tumor volume through improved CD8 infiltration, which improved overall survival and risk of metastasis when combined with PD-L1 blockade[4]. Similarly, depleting hyaluronan, a large polysaccharide component of the tumor ECM, can improve therapy response and reduce tumor burden in preclinical studies[6–8]. These studies indicate that the tumor ECM may have a direct

[1]Queen Mary University of London, Barts Cancer Institute, John Vane Science Centre, London EC1M 6BQ, UK. [2]Department of Biomedicine and Division of Medical Oncology, University Hospital Basel, Hebelstrasse 20, 4031 Basel, Switzerland. [3]Department of Women and Children Health, University of Padova and Fondazione Istituto di Ricerca Pediatrica Città della Speranza, Corso Stati Uniti 4, 35127 Padova, Italy. [4]UCB Pharma Ltd, 208 Bath Road, Slough, Berkshire SL1 3WE, UK. [5]These authors contributed equally: E. H. Puttock, E. J. Tyler. ✉e-mail: o.pearce@qmul.ac.uk

role in inhibiting immunity, which could occur through a physical blockade of cell movement and/or through receptor-ligand interactions. Whether the tumor ECM can directly educate the phenotype of infiltrating immune cells is not well understood but could prove important to our understanding of tumor immunity and improve immunotherapy efficacy.

In our previous work, we identified a specific tumor ECM signature that associated with disease progression and immunosuppression, identified in high-grade serous ovarian cancer (HGSOC)[1], and shared by 12 other solid carcinomas, including those with limited treatment options such as pancreatic, lung, and esophageal cancers. This ECM signature associated with immunosuppressive cell phenotypes, but did not positively correlate with cytotoxic immune cell phenotypes, and negatively associated with CD8 T cells[9]. Building on these studies, we applied computational algorithms CIBERSORTx and xCell to these previously gathered datasets[1] to further look at how specific ECM molecules associate with immune cell infiltrate. This analysis identified a population of tumor-associated macrophages (TAMs) associated with five ECM molecules that are predictive of poor prognosis, and have been identified previously in signatures that associate with immunosuppression, poor prognosis, and failure of immunotherapy response[2]. These macrophages were characterized by the gene signature for M0 macrophages that are used in CIBERSORTx and xCell. Originally the signature for M0 macrophages was used to define a non-activated macrophage phenotype, however, more recent studies indicate that the signature is also a feature of a poorly characterized macrophage population that associates with poor prognosis in hepatocellular[10], sarcoma[11], breast[12,13], and lung[14] cancers. TAMs likely exist as a spectrum of subtypes influenced by their interaction with malignant and immune cells and secreted factors during infiltration into the tumor tissue[15,16]. At the poles of TAM polarization, classically activated, or M1-like, macrophages support inflammation and associate with better outcomes, whereas alternatively activated, or M2-like macrophages suppress anti-tumor immunity. These macrophage phenotypes can be generated through cytokines released within the local microenvironment. For example, macrophages that exhibit an M1-like phenotype are stimulated through interferon-gamma (IFN-γ), whereas macrophages that exhibit an M2-like phenotype are stimulated through interleukins (IL) such as IL-4 and IL-13. Whether the tumor ECM is also a factor in the generation of TAM populations is not well understood. To answer this question, we made a decellularized tissue model of omental metastasis from whole patient samples, which comprises a heterogeneous ECM derived from a variety of tumor microenvironment (TME) cell types, where we found monocytes differentiate into TAMs that overlap with the M0 and M2 signatures. These ECM-derived TAMs have gene programs associated with T cell activation and tissue remodeling. This study demonstrates that tumor ECM can directly educate TAMs found in ovarian cancer tissues that associate with poor prognosis and suggests that strategies that target ECM may also alter immune cell phenotype as well as their infiltration within the tumor microenvironment.

## Results

### An HGSOC macrophage subtype associates with a prognostic tumor ECM signature

Several bioinformatics studies have connected immune cell infiltration with the composition of the tumor ECM[1,2,17]. To investigate changes that occur to immune cell phenotypes in ovarian metastasis we used computational algorithms CIBERSORT, CIBERSORTx, and xCell[18–21] to predict and deconvolute immune cell abundances in tumor tissue from patient biopsies (Fig. 1A). To investigate ECM molecules differentially expressed and associated with immune cells we compared immune cell gene signatures against ECM composition within our previously published dataset of omental metastasis of HGSOC samples where transcriptomic, proteomic, and immune cell count data were

available[1] (Fig. 1, Supplementary Fig. 1, and Supplementary Methods). As anticipated, deconvolution of the bulk RNA-sequencing (RNAseq) dataset using analytical tools CIBERSORTx and xCell (which were designed to analyze RNAseq data) accurately estimated the abundance of CD8+ ($p < 0.001$ and $p = 0.0001$, respectively), and CD68+ ($p < 0.05$ and $p < 0.01$, respectively) cells, providing further confirmation that transcriptional expression of macrophage markers correlates with the immune cell counts by IHC (Supplementary Fig. 1D, E). We could not use IHC immune cell counts to verify the presence of all the immune subtypes, as we compared a restricted panel of only six IHC markers against the 22 or 34 immune cell subtypes detected by CIBERSORTx and xCell. However, the computed subtypes are well-validated in the literature[18,19,21] (Supplementary Dataset 1). CIBERSORT, an earlier version designed to analyze microarray data, did not accurately estimate abundances except CD8+ and was not used further. We next compared the immune signatures generated by both CIBERSORTx and xCell against a previously reported tumor ECM signature and measure of tumor desmoplasia[1] in the same cohort of patients identifying several correlating immune cell types including macrophages, B cells, several T cell subsets, dendritic cells, basophils, and neutrophils, which together confirmed our previous results using a less comprehensive set of immune cell signatures (Supplementary Fig. 2A, B). Of note, macrophages were associated with these signatures, and the majority of intra-tumoral macrophages are thought to exhibit an M2-like phenotype which correlates with poor prognosis in several malignancies, including ovarian cancer[22,23]. We identified M2 macrophages as a predominant signature present in all our HGSOC patient samples (Supplementary Fig. 1A–C). However, we did not find a significant association for alternatively activated 'M2-like' macrophages with the extent of disease (extent of disease is a measure of tumor cell and desmoplasia percentage area within a tissue section) (Supplementary Fig. 2A, B), an observation which has been reported previously[24,25], which may result from heterogeneity between patient samples. Extent of disease and tumor ECM associated most strongly with the macrophage signatures described by CIBERSORTx 'M0 macrophage' and 'Macrophage' xCell signature, both of which are derived from the Immune Response In Silico (IRIS) gene expression dataset for macrophages after 7 days of differentiation from monocytes[18,19,26] (Supplementary Fig. 2A, B). The CIBERSORTx M0 macrophage gene signature has been more recently associated with tumor promotion and poor prognosis in multiple cancers[10–14], although it is unclear if the M0 signature describes a distinct population of macrophages or (more likely) multiple populations of macrophages, which does not fit well with classical and non-classical macrophage signatures. The M0 population has been previously reported in murine HGSOC mouse models[27]; however, to our knowledge, this is the first reported observation of M0 macrophages within the TME of patients with HGSOC.

Next, we wanted to investigate correlations of immune cell populations with specific ECM molecules. To identify these significant immune-ECM associations, we integrated sample matched CIBERSORTx and xCell immune signatures against gene (Supplementary Fig. 3) and protein (Fig. 1B–E) expression values for ECM molecules found in the matrisome database[28], using Spearman's correlation (Supplementary Datasets 2 and 3, respectively). 16 ECM molecules were significantly positively associated with immune cell signatures at gene and protein levels, including macrophages, T cells, and B cells (Supplementary Fig. 4). Five ECM molecules, fibronectin (FN1), versican (VCAN), matrix remodeling associated 5 (MXRA5), collagen 11 (COL11A1), and secreted fizzled related protein 2 (SFRP2) were consistently associated with CIBERSORTx M0 Macrophage and xCell Macrophage signatures (Fig. 1F, Supplementary Dataset 4). Individually these five molecules associated with several immune cell types (Fig. 1D, E), but only the association with M0 macrophages was shared by all five molecules across both the CIBERSORTx and xCell approaches (Fig. 1F). To explore the association of these five M0 macrophage-

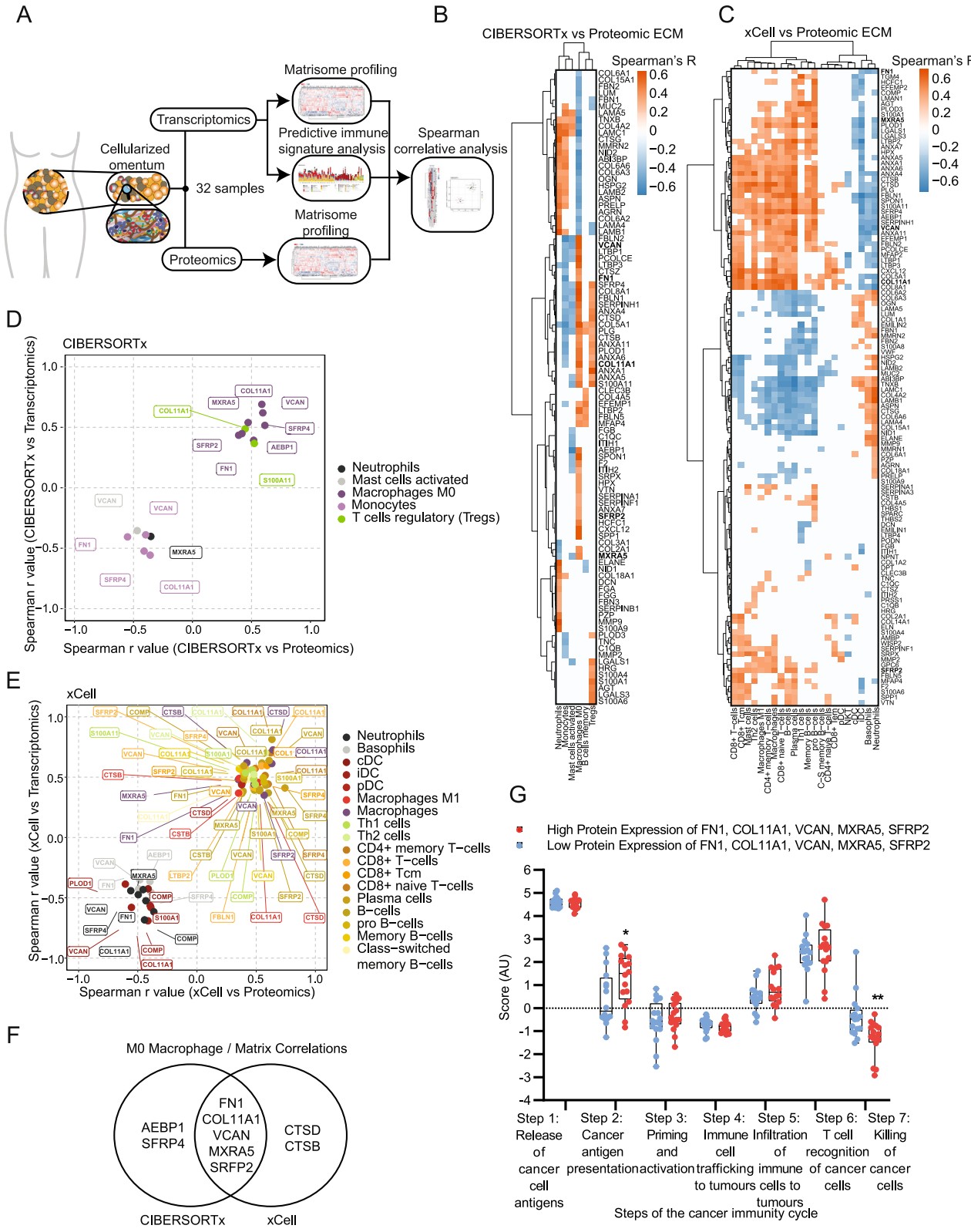

associated ECM proteins with the immune microenvironment in cancer, we tested whether the signature associated with the cancer immunity cycle (Fig. 1G), a series of seven step-wise events that help inform how an immune response can be activated against an established tumor[29]. To do this, we used TIP[30] (Tracking Tumor Immunophenotype), which showed a significant reduction in cancer killing activity (step 7 of the cancer immunity cycle, Fig. 1G) suggesting that

this ECM signature may be related to the presence of immunosuppressive phenotypes (Fig. 1G). TIP analysis also showed an increase in step 2, cancer antigen presentation, which was not recapitulated in TIP analyses of the individual ECM proteins, suggesting that all five ECM proteins are contributing to this difference, so does not seem to be one particular protein but rather the pattern (Fig. 1G, Supplementary Fig. 5). No further significant differences between the other five steps

**Fig. 1 | M0 macrophages are enriched in highly diseased HGSOC samples and significantly correlate with a matrisome signature. A** Schematic of bioinformatics pipeline. **B** Heatmap of ECM proteins significantly associated with immune cells from CIBERSORTx analysis. Spearman's r values with two-sided alternative hypothesis testing representing correlation between immune abundances and protein expression levels are plotted according to the color scale, with positive correlations in red and negative in blue. White colored associations are insignificant with a correlation $p > 0.05$, $N = 32$ HGSOC samples. Ordered by unsupervised clustering. **C** Heatmap of ECM proteins significantly associated with significant immune cells from xCell analysis. Spearman's r values with two-sided alternative hypothesis testing representing correlation between immune abundances and protein expression levels are plotted according to the color scale, with positive correlations in red and negative in blue. Cream-colored associations are insignificant with a correlation $p > 0.05$, $N = 32$ HGSOC samples. Ordered by unsupervised

clustering. **D, E** Scatterplot of significant protein and gene correlations with associated immune cell types using **D** CIBERSORTx and **E** xCell in HGSOC. Spearman's regression analysis with two-sided alternative hypothesis testing was completed to generate plotted r values. Points are colored based on their associated immune cell type as depicted in the key. **F** Venn diagram showing overlap of ECM molecules significantly associated with CIBERSORTx M0 macrophage signature and xCell macrophage signature. **G** Boxplot of predicted tumor immune phenotype stage scores by TIP, with HGSOC samples separated by high (red) and low (blue) protein expression levels of FN1, VCAN, MXRA5, COL11A1, SFRP2. Scores are based upon expression of signature genes and represent activity levels. Within each box, the center line denotes the median value (50th percentile) while the box extends from the 25th to the 75th percentile; whiskers mark the maximum and minimum values. Two-way ANOVA significance between each group is presented as **$p = 0.005$ and *$p = 0.0204$. $N = 32$ HGSOC samples (16 low and 16 high).

of the cycle were observed (Fig. 1G). Laser capture microscopy of tumor and stroma areas from HGSOC tissues from two patients in the same cohort as the above analysis showed that all five matrisome molecules associated with M0 macrophage phenotype are found predominately in the stroma (Supplementary Fig. 6, Supplementary Video 1). Finally, we used an online resource[31] (https://kmplot.com/analysis/) to determine whether the ECM signature that associates with M0 macrophages associated with overall survival (OS), where we found a significant decrease in the median OS of ovarian cancer patients (OS = 28.25 months) compared to the low expression cohort (OS = 45.63 months; Supplementary Fig. 7A), with a greater hazard ratio (HR = 1.56) than the expression of any gene alone, except COL11A1 or SFRP2 (HR = 1.78 and HR = 1.66, respectively, Supplementary Fig. 8). Further analysis using the KM Plotter for pan-cancer across 19 carcinomas found 12 cancers (including ovarian cancer) to have reduced patient OS with high expression of the ECM signature associated with M0 macrophages (Supplementary Fig. 7B). Taken together, we found that the composition of the tumor ECM which correlates with extent of disease associates most strongly with a M0 macrophage subtype which is poorly characterized in HGSOC, and five tumor ECM proteins which strongly associated with this macrophage subtype showed a reduced cancer cell killing signature and poor prognosis across 12 cancer types.

### Generation of a decellularized tissue model of ovarian cancer metastases

Having found that tumor ECM composition associates with a poor prognostic ECM signature associated with M0 TAMs (Fig. 1), we next evaluated whether this association resulted from the composition of ECM, using a decellularized tissue model of tumor ECM, made from a tissue library of 39 human omental ovarian metastatic tumor samples (Supplementary Dataset 5, Fig. 2A). Prior to decellularization, each tissue sample was characterized to assess the extent of disease, and number of infiltrating immune cells (Supplementary Fig. 9). Following IHC characterization, each tissue was cut to a uniform thickness (300 μm) using vibratome sectioning, and decellularized using a process of cell lysis, lipid extraction, and DNA degradation buffers as described in the methodology section. Tissues were effectively decellularized, shown by nucleic acid content and nuclei content as analyzed by spectrophotometer and IHC, respectively (Fig. 2B, C, Supplementary Fig. 10A, B). We next used IHC to assess the content of four ECM molecules (collagen 1 (COL1A1), fibronectin (FN1), versican (VCAN), cathepsin-B (CTSB), and one glycosaminoglycan chondroitin sulfate (CS), which were selected based on their high level of expression in cancer tissues[1] and/or their association with the macrophage phenotype as described above (Fig. 1). Using Definiens® digital software analysis, we compared the five molecules expression levels between matched cellularized and decellularized tissues and found that the expression levels were comparable for four out of the five (Fig. 2D, Supplementary Fig. 11A). CTSB content appeared to increase

after decellularization, resulting from removal of the cellular component, as the presence of hemotoxylin in the cellularized tissue was overlapping and masking the intensity of CTSB (Supplementary Fig. 11A). The expression of chondroitin sulfate (CS), which decorates the surface of many proteoglycans and comprises a major constituent of the post-translational environment, was also maintained after decellularization (Fig. 2D). We next tested whether decellularization affected the ECM architecture using scanning electron microscopy (SEM) (Fig. 2E, Supplementary Fig. 11B), and compared ECM fiber diameter and alignment between matched cellularized and decellularized tissues as described in the methodology section. We also performed Masson's trichrome stain to measure collagen ECM architecture and used the TWOMBLI[32] plugin (within ImageJ) to quantify differences in alignment (Fig. 2H, I, Supplementary Fig. 12). Taken together, we observed no change between cellularized and decellularized tissues (Fig. 2E–I). Both low and high disease samples displayed highly aligned fibers, consistent with the literature demonstrating aligned collagen fibers in HGSOC ovarian tissue using second harmonic generation imaging[33,34]. Finally, we tested whether the decellularized tissues were a biocompatible material for cell culture. Using immunofluorescence (IF), we found monocytes/ macrophages were viable for a period of at least 14 days (Fig. 2J) and flow cytometry revealed macrophages could be isolated from decellularized tissues and analyzed for marker expression (Fig. 2K). Flow cytometry analysis detected an increased level of cell death in cells cultured on decellularized tissue compared to tissue culture plastic (TCP) (approximately 5–10% more cell death, Fig. 2K, Supplementary Fig. 11C), however, this was due to cell death during the removal of the cells from the tissue as cells adhered firmly in the decellularized tissue. We next optimized the cell recovery after finding a significant effect on cell marker expression by flow cytometry between macrophages isolated from TCP versus decellularized tissue when using enzyme-based dissociation solutions, which could be negated by using an enzyme-free buffer (Supplementary Fig. 13); this effect has similarly been reported previously[35]. Altogether, we concluded that the decellularization process maintained the ECM architecture with minimal effect on the quantity of major ECM components, and decellularized tissues were a suitable platform for cell culture, where cells could be visualized on the tissue or removed as single-cell suspensions for further assays or analysis.

### The M0 associated ECM signature is also a feature within the present tissue library and positively associated with TAM infiltration

We investigated whether the association of ECM composition and TAM phenotype seen in the published analysis cohort[1] (Fig. 1) was a feature in the library of ovarian cancer metastatic tissues used here to make the decellularized tissue platform (prepared in Fig. 2), through integration of proteomics and IHC immune cell counts analyzed by QuPath (Fig. 3A, immune cell counts per sample are shown in Supplementary

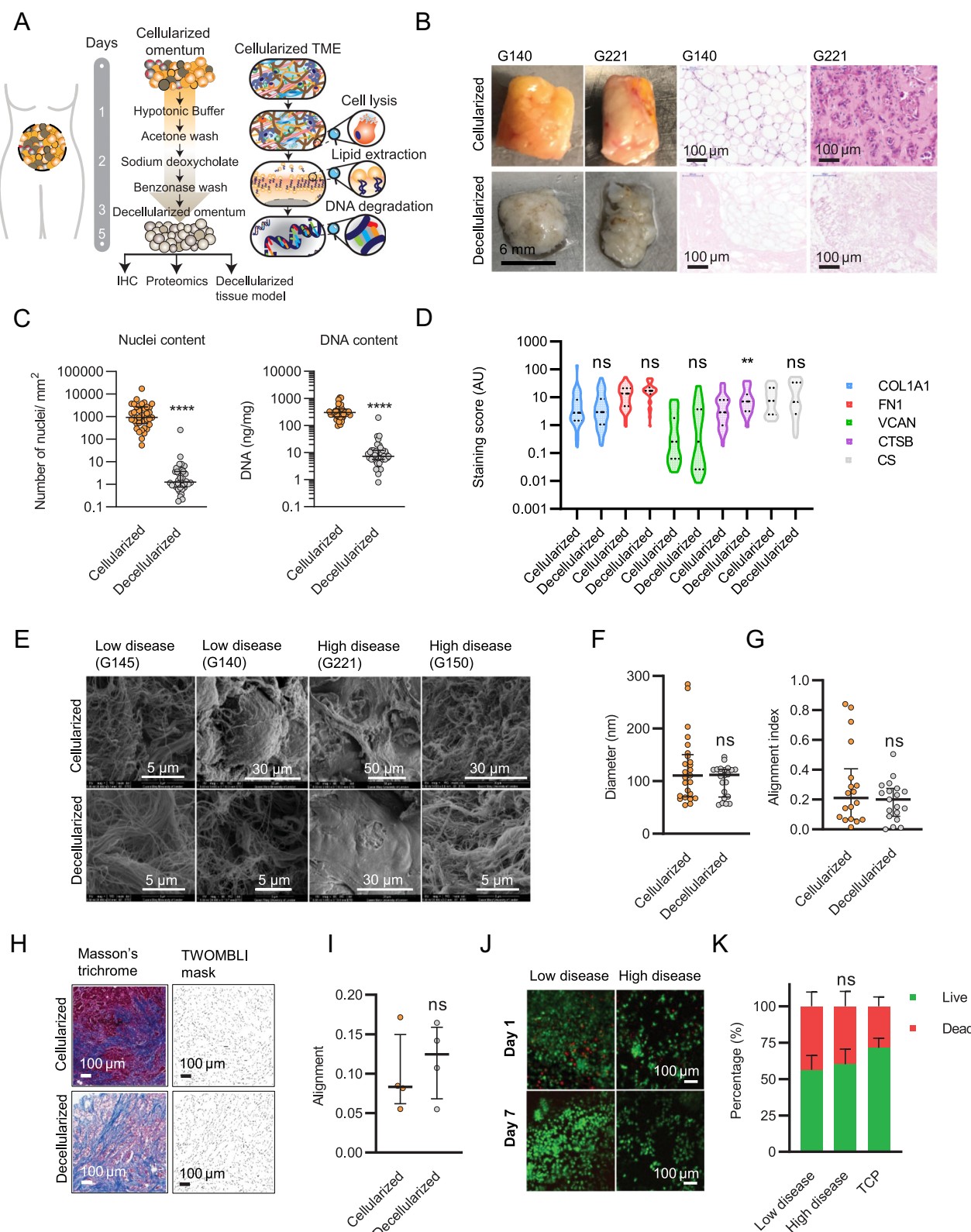

Fig. 15). We characterized the ECM of the decellularized samples using ECM-focused proteomics[36] (Supplementary Fig. 16A, and Supplementary Dataset 6). The proteomics data indicated collagens constituted the largest proportion of the ECM by abundance (35%), followed by glycoproteins (25%) and proteoglycans (25%), ECM-regulators (8%), ECM-affiliated proteins (5%), and finally ECM secreted factors (1%) (Supplementary Fig. 16A). Hierarchical clustering separated the

samples by their proteomic ECM composition which identified five ECM composition groups (ECGs) (Fig. 3B). ECGs were compared to the disease score (Supplementary Fig. 16B, Supplementary Dataset 7). The disease score is a digital histopathology method we described previously[1], that measures extent of disease present within a sample based on the area of tumor (PAX8 positivity) and stroma (hematoxylin staining of areas where there is stromal remodeling or existing

**Fig. 2 | A decellularized tissue model of ovarian cancer metastasis. A** Schematic of the decellularization procedure and tissue processing. **B** Fresh tissue biopsies taken from ovarian cancer patients. Histologically uninvolved samples from benign tumors were used as part of the group of low disease tissues. H&E staining of sectioned ovarian cancer samples (G140, G221). Tissue biopsies, Scale bar = 6 mm. H&E, Scale bar = 100 μm. **C** Nuclei and nucleic acid content in cellularized and decellularized tissue samples. Data are presented as median values +/− inter-quartile range (IQR). Two-tailed unpaired t test. Nuclei ****$p = 0.000085$; Nucleic acid content ****$p = 1.198 \times 10^{-13}$. $N = 39$ each. **D** IHC staining analysis using Definiens® digital image software for matrix molecules in cellularized and decellularized samples. Violin plot in which the center line denotes the median value (50th percentile) and dashed lines denote the 25th and 75th percentile. Mixed-effects analysis, with Sidak's multiple comparisons test, **$p = 0.0096$. $N = 39$ per stain. **E** Representative Scanning Electron Microscopy (SEM) images for cellularized and decellularized samples. $N = 6$ each. **F** Quantitative fiber diameter analysis from SEM images. Three to six fields of view were chosen at random in cellularized and matched decellularized tissue micrographs. Data are presented as median values +/− IQR. Two-tailed unpaired t test. $N = 6$ each. Fiber diameter angles were recorded (minimum 30 fibers quantified per field of view). **G** Quantitative fiber alignment (alignment index) from SEM images. The alignment index ranges from 0 to 1, with 1 indicating perfect alignment, and 0 indicating disorganized fibers (as described in the methodology). Data are presented as median values +/− IQR. Two-tailed unpaired t test. $N = 6$ each. **H** Representative images of cellularized and decellularized tissue stained by Masson's trichrome stain and representative images of the TWOMBLI mask generated for alignment analysis. Four fields of view were chosen at random. Scale bar = 100 μm. **I** TWOMBLI analysis showing alignment of collagen fibers stained in blue. Data are presented as median values +/− IQR. Two-tailed unpaired t test. $N = 4$ tissues. **J** Representative live (green)/dead (red) immunofluorescence (IF) images from decellularized tissue model cultures using monocytes/macrophages at day 1 and 7. $N = 3$. Scale bar = 100 μm. **K** Flow cytometry analysis of viable macrophages collected after 14 days of culture from low disease samples, high disease samples and tissue culture plastic (TCP). Data are presented as mean values +/− standard deviation (SD). Two-way ANOVA with Sidak's multiple comparisons test. $N = 3$.

---

stroma). Previously we found disease score to be associated with an increase in immune infiltration, tissue stiffness, and ECM remodeling. In the sample library presented here (Supplementary Dataset 5) we found samples split into two groups corresponding to low disease score (ECG1 and 2), where samples primarily consisted of adipocytes, or high disease score groups (ECG3, 4, and 5), where samples contained high levels of PAX8+ cells and stroma (Fig. 3B, Supplementary Fig. 16B). There was a significant difference in the disease scores between ECG1-2 when compared against ECG3, 4, or 5, indicating the change in ECM composition that occurs with extent of disease[1]. Tissue samples in ECG1-2 were mostly adipose tissue showing only a low/no level of tumor and stroma as observed from their disease scoring (Supplementary Fig. 9D). We analyzed the difference in ECM composition between ECGs 3, 4, and 5, all of which contain samples with significant levels of tumor and stroma and similar disease scores that are high in comparison to ECG1-2 (Supplementary Fig. 16B). The overall expression of major ECM categories between ECGs 3–5 appeared to be similar, with the exception of ECG4, where an expansion of ECM-glycoproteins was observed (Supplementary Fig. 16C). Next, we analyzed the differentially expressed (DE) proteins between each ECG and found each had a different pattern of ECM protein expression resulting from changes in the abundance of specific molecules (Fig. 3C). We explored the ECM signature associated with M0 macrophages (Fig. 1F) across the ECG groups. Individually, the ECM signature molecules tended to be highly expressed in ECG3 and ECG5 and appeared highest in ECG5, with the exception of COL11A1 which did not correlate with a specific ECG group(s) and SFRP2 which was not detected in this proteomic dataset. In particular, FN1 expression was significantly higher in ECG5 (Fig. 3D). We also compared the ECM signature associated with M0 macrophages (Fig. 1F) across the five ECG groups finding ECG5 was enriched for the signature (Fig. 3D, right-hand panel). These data indicated ECM composition can vary across ovarian patient samples, an observation we had previously noted from transcriptomic data for primary HGSOC within The Cancer Genome Atlas[1]. Taken together, tumors with similar disease scores can have different ECM compositions. To explore the localization of these matrisome molecules, we integrated the above proteomics data with an analysis which defined adipose, stroma and tumor areas in the same tissues using IHC data (Supplementary Fig. 17). The expression of FN1 and VCAN was significantly positively associated with the percentage of stroma area, but not tumor area, and were significantly negatively associated with the percentage of adipose area, indicating that, for the most part, these ECM molecules were being expressed by cells found within the stroma (Supplementary Fig. 17). Subsequently, proteomic data of cell-derived matrices (CDMs) from patient-derived omental fibroblasts and metastatic HGSOC malignant cell lines were measured to explore the expression of these matrisome molecules in in vitro cultures. In line with our previous results, four of the matrisome molecules associated with M0 macrophages (FN1, VCAN, COL11A1, MXRA5) were more highly expressed by patient-derived omental fibroblasts than malignant patient-derived HGSOC cell lines (Supplementary Fig. 18). We wondered if the ECM patterns seen in our decellularized ovarian metastatic samples (Fig. 3C) may correlate with the immune infiltrate, as we had found from our informatics analysis (Fig. 1). We then integrated ECM proteomics data (Fig. 3C) with immune cell counts determined from IHC and analyzed by QuPath (Supplementary Figs. 9 and 14, Supplementary Dataset 8). Immune cell counts were evaluated across ECGs (Fig. 3E, F). Low disease omental samples (ECG1-2) composed almost entirely of adipose tissue (Supplementary Fig. 9D) had ECM compositions comparable to normal tissues[36] and had low immune cell abundances (Fig. 3E, F). By contrast, the level of immune infiltrates changed significantly between ECGs 3 < 4 < 5 (Fig. 3E). We found that macrophages (CD68+) were the dominant immune cell type in all ECGs, although in ECG3-5 they were significantly more of them and they were the most abundant immune cell type, being highest in ECG5 (Fig. 3E). These data confirmed the earlier analysis we had performed on a separate set of tissues[1] (Fig. 1). The number of CD4+ and CD8+ cells varied between ECG3-5, with CD8+ cells highest in ECG5 and CD4+, FOXP3+, and CD20+ highest in ECG4. We calculated z-scores from IHC cell count data and categorized by ECGs (Fig. 3F), which revealed the trend of increasing CD8+ and CD68+ cell counts between the three groups ECG3 < ECG4 < ECG5. We evaluated if the presence of tumor cells (PAX8+ cells) may also correlate with immune cell counts between ECG3-5, however, we found no correlation when looking at tumor samples (Supplementary Fig. 16D–G). Together, these data identified an association between the tumor ECM composition and immune infiltrate that confirmed our findings from Fig. 1.

### In vitro ECM educated TAMs share gene profiles with M0 TAMs found in human tissues

Using the decellularized tissue model, we explored whether the tumor ECM could directly educate monocytes to differentiate into the M0 TAM phenotype found in ovarian metastases (Figs. 1 and 3). We cultured monocytes derived from healthy donor peripheral blood mononuclear cells (PBMCs) on decellularized tissues from our tissue library (Fig. 2 and Supplementary Fig. 9) that represented ECM compositions of low disease (LD) tissues (ECG1-2, four patient tissues) or high disease (HD) tissues (ECG3 & 5, four patient tissues) (Fig. 4A), and after 14 days, macrophages were collected and bulk RNA-sequencing (RNAseq) was performed (Fig. 4B, Supplementary Fig. 19A, Supplementary Dataset 9). Transcriptomic profiles grouped samples based on which decellularized tissue they were cultured on (Fig. 4B, Supplementary Fig. 19B); macrophages cultured on their respective decellularized tissue type (either HD or LD) generally clustered into

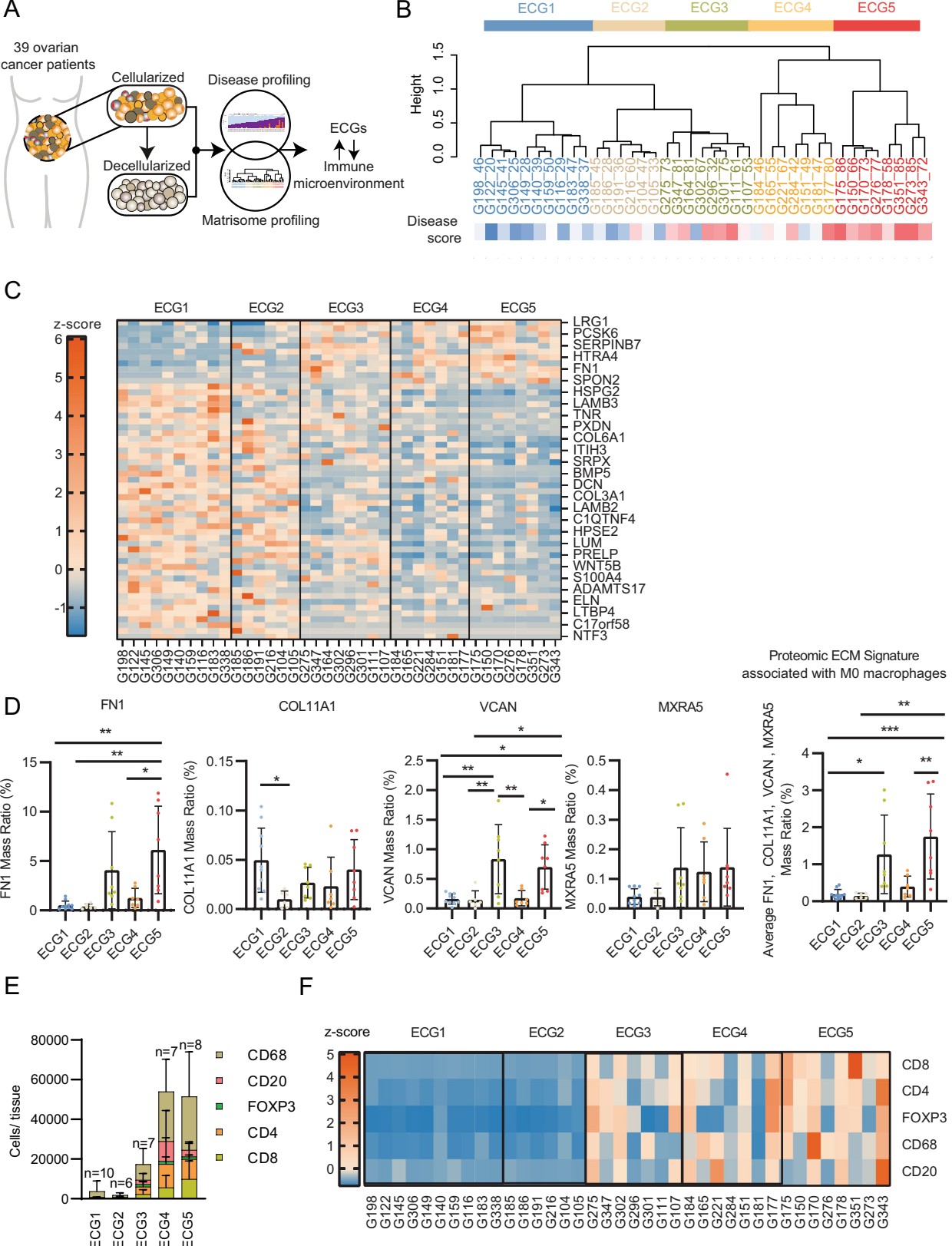

their respective group (i.e., macrophages cultured on HD tissue clustered with other samples of that tissue type). Differential gene expression analysis identified a total of 3613 genes between macrophages cultured on the HD tissue (ECG3 & 5, 1839 upregulated genes) and macrophages educated on the LD tissue (ECG1 & 2, 1774 upregulated genes) (Supplementary Fig. 19C, D). CIBERSORTx analysis of DE

genes revealed that macrophages educated by the HD or LD decellularized ECM shared gene profiles with the M0 macrophage subtype (Fig. 4C), leading us to term these as extracellular matrix-educated macrophages (MAMs). Gene analysis revealed HD MAMs contained significantly more of the M0 macrophage subtype and shared signatures with M2-like macrophages, when compared to LD MAMs

**Fig. 3 | ECM composition correlates with the immune cell landscape.**
**A** Schematic showing integration analysis using the disease score (disease profiling) and ECM proteomics (matrisome profiling) of ovarian metastatic samples to define the ECM composition and explore the synergy with ECM composition and the immune cell landscape. **B** Hierarchal unsupervised clustering (ward.D2 method) separated the tissue samples into five groups, based on the samples ECM protein expression, that we have termed as ECM composition groups (ECG) 1–5. Labels: tissue ID _disease score. Disease score (DS) displayed as heatmap from low disease score (blue) to high (red). $N = 39$ samples. **C** Heat map using row z-scores of positively and negatively regulated matrisome proteins, columns grouped by ECG1-5. Protein names illustrated for every other row. Full list in source data. $N = 39$ samples. **D** Bar plots of protein expression (proteomics) for FN1, VCAN,

COL11A1 and MXRA5 in ECG1-5. Data are presented as mean values +/− SD. One-way ANOVA with Tukey's post-hoc test, significance between each group is presented as $**p < 0.01$ and $*p < 0.05$. ECG1 $N = 10$ samples; ECG2 $N = 6$ samples; ECG3 $N = 8$ samples; ECG4 $N = 7$ samples; ECG5 $N = 8$ samples. **E** Immune cells were detected by IHC and analyzed by QuPath; mean number of immune cells/number of tissues between ECG1-5. Data are presented as mean values +/− SD. ECG1 $N = 10$ samples; ECG2 $N = 6$ samples; ECG3 $N = 7$ samples (G164 not included as not enough tissue for this analysis); ECG4 $N = 7$ samples; ECG5 $N = 8$ samples. **F** Row z-scores of IHC immune cells/mm². ECG1 $N = 10$ samples; ECG2 $N = 6$ samples; ECG3 $N = 7$ samples (G164 not included as not enough tissue for this analysis); ECG4 $N = 7$ samples; ECG5 $N = 8$ samples.

(Fig. 4D, E, Supplementary Fig. 20). Conversely, LD MAMs showed similarities to an M1-like phenotype, when compared to HD MAMs (Fig. 4D, E). Subsequently, we performed an analysis to relate the significantly upregulated genes in HD or LD MAMs to a spectrum-model[37] of macrophage activation generated by stimulating human CD14+ monocyte-derived macrophages using 29 discrete stimuli that revealed 49 co-expressed gene modules (Supplementary Fig. 21). HD MAMs significantly positively associated with modules that occur upon in vitro activation with stimuli linked to fatty acids (palmitic acid, oleic acid, linoleic acid; Supplementary Fig. 21B). Whereas LD MAMs significantly positively correlate with modules that occur upon in vitro activation with stimuli linked to LPS and to chronic inflammation (TNF, prostaglandin E2, TLR2-ligand; Supplementary Fig. 21B). These data indicate that both MAM phenotypes strongly associated with stimuli from a spectrum extended beyond M1 versus M2 polarization. Taken together, we conclude that HD MAMs share transcriptomic similarities to the macrophage phenotype seen in HGSOC tissues, and therefore we conclude the ECM is capable of stimulating the TAM population seen in HGSOC.

## HD MAMs have an immunoregulatory and tissue remodeling phenotype

We next investigated the macrophage phenotypic transition by characterizing cell surface markers in MAMs educated by HD (ECG3&5) or LD (ECG1 & 2) decellularized tissue. We used flow cytometry to compare the expression of macrophage markers including CD14, CD11b, CD45, CD163, CD206, CD86, CD38, CD209, CD47, CD36, and CD204, between HD and LD MAMs (Fig. 4F, Supplementary Fig. 22). Transmembrane receptors including alternatively activated macrophage marker, CD163, were significantly upregulated on HD MAMs, while mannose receptor, CD209, was upregulated on LD MAMs (Fig. 4G). These data confirmed our observation from the CIBERSORTx transcriptomic analysis (Fig. 4C–E) that the matured macrophages were not predominantly polarized into the classically defined 'M1' or 'M2'-like states. Next, we looked at the total and the top 30 DE genes that associated with HD MAMs (Fig. 4H). We selected some of the top DE genes including transmembrane receptors, immunoregulatory soluble factors, ECM-collagens, ECM-regulators, transcription factors, scavenger receptors like CD36, C-X-C motif chemokine ligand 5 (CXCL5), arginase (ARG1) and a number of matrix metalloproteinases (MMPs) (Fig. 4I). The patterns of upregulated genes indicated HD MAMs may have immunoregulatory and ECM remodeling activity. Since several soluble factors seemed to be expressed by HD MAMs, we used a multiplex assay (LEGENDplex™) to characterize a panel of macrophage-secreted factors at the protein level (Supplementary Fig. 23, Supplementary Dataset 10). This analysis revealed chemokine CXCL5, a chemoattractant of T cells, and a member of the M0 CIBERSORTx signature, was significantly upregulated by HD MAMs, at gene and protein level (Fig. 4I, J). Other secreted factors including CCL2 which was significantly upregulated by LD MAMs, and CXCL1, which showed no change between HD or LD MAMs was detected at very low levels (Fig. 4K, Supplementary Fig. 23). To explore which

biological processes may be altered in HD MAMs, we used weighted gene correlation network analysis (WGCNA), which identified 13 coordinately expressed gene programs associated with distinct biological pathways (Fig. 5A, Supplementary Dataset 11). Hierarchal cluster analysis separated the gene programs by their similarity (Fig. 5B). Gene programs such as clusters 8, 9, and 10 were significantly upregulated in HD MAMs, while gene programs contained within clusters 1, 2, and 12 and other WGCNA programs were downregulated (Fig. 5C). HD MAMs upregulated programs (clusters 8, 9, and 10) that were distinctly enriched for integrin receptors (blood coagulation pathway), Toll-like receptor signaling, ECM organization and ECM disassembly and leukocyte cell adhesion, while downregulated gene programs (clusters 1, 2, 4, and 12) included pathways of the inflammatory response such as MHC class II antigen processing and presentation, IFN-γ signaling, and defense response (Fig. 5D). This indicated HD MAMs were associating with an integrin-ECM interaction and were involved in ECM remodeling via both construction (ECM organization) and deconstruction (ECM disassembly). We built a HD MAM gene signature based on DE genes and TAM-like markers from the transcriptomic dataset (Fig. 5E) and associated this with ECM receptors, collagen degradation molecules, and classic macrophage activation genes which further associated HD MAMs with a tissue remodeling phenotype (Fig. 5F). Taken together, these data describe the HD MAM phenotype as having immunoregulatory and tissue-remodeling capabilities, which are also predicted characteristics of the M0 macrophage phenotype. This suggests these same macrophage processes may also be driven by HD ECM in vivo. We next focused on whether the HD MAMs may have immunoregulatory properties as predicted from the transcriptomic profiles.

## HD MAMs have reduced phagocytic function and alter T cell activation

HD MAMs are predicted to be immunoregulatory based on transcriptomic profiles and their secretome (Figs. 4 and 5). To evaluate MAM immune function, we first performed a phagocytosis assay; phagocytosis being an intrinsic function of macrophages[38,39]. Briefly, MAMs were mixed with CellTrace Yellow stained (CTY+) K562 cells and flow cytometry was used to measure the number of phagocytic events (i.e., CTY+ MAMs). HD MAMs had a reduced phagocytic function compared to LD MAMs (Fig. 6A). We thought this reduced phagocytosis may be due to the difference in expression of some transmembrane receptors such as CD209 (which was expressed lower in HD MAMs at gene and protein level) (Fig. 4G, I). CD209 has previously associated with macrophage particle uptake[40]; however, we found no difference in the MAMs phagocytic capacity when selecting for CD209+ or CD206+ macrophages (Fig. 6B). Because we previously showed that HD MAMs have upregulated toll-like receptor signaling (Fig. 5D) which can promote production of pro-inflammatory cytokines and chemokines (Fig. 4I), we wondered whether HD MAMs may affect T cell function. CD3+ T cells were cultured with low-level CD3/CD28 stimulation with HD or LD MAMs and in parallel, we cultured CD3+ T cells with HD or LD MAMs conditioned media to test whether

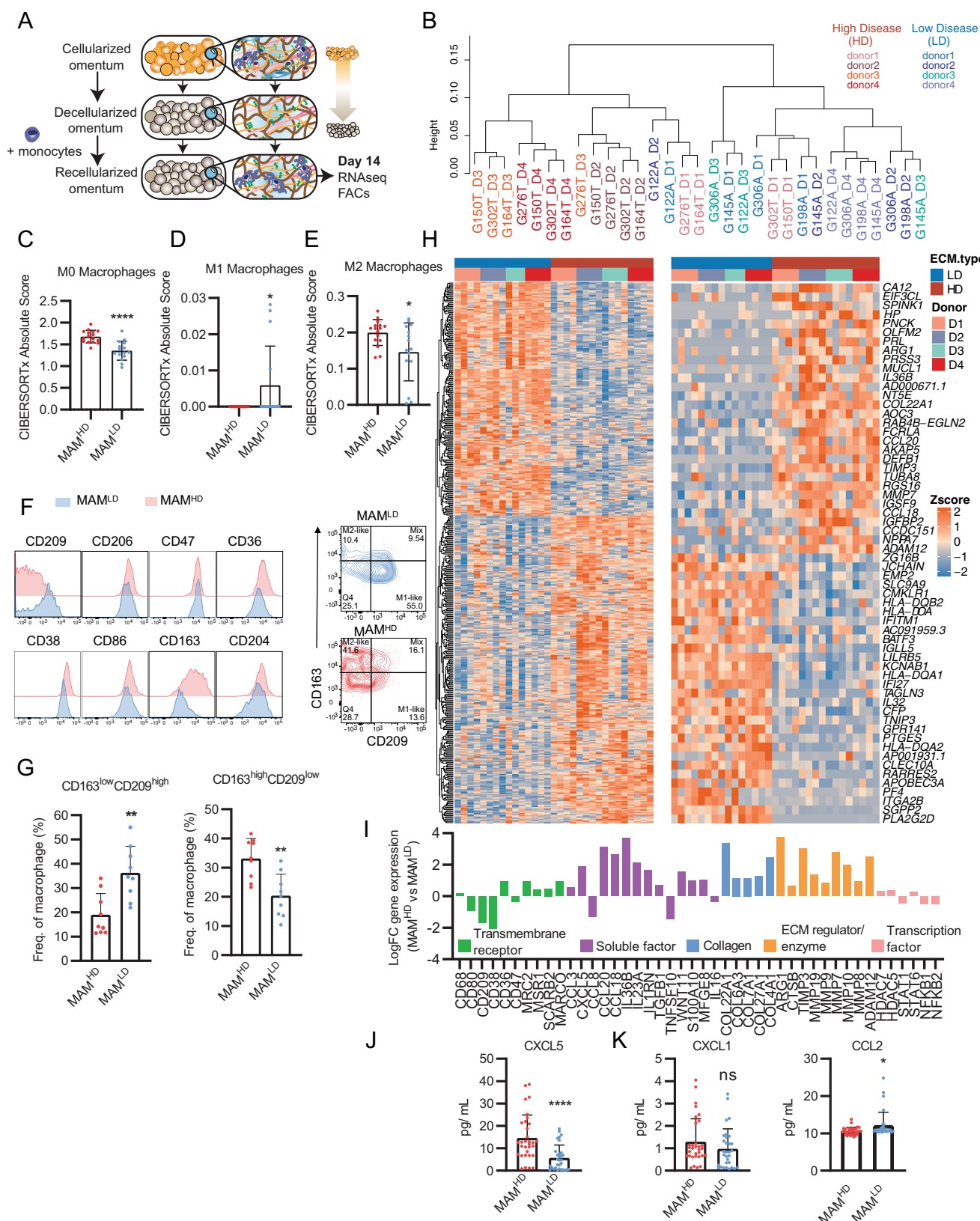

these macrophages directly or indirectly (macrophage-secreted factors) modulated T cell phenotype and proliferation (Fig. 6C). HD MAMs induced inhibitory receptor expression of LAG3 on CD3$^+$ T cells, compared with LD MAMs and control (Fig. 6D). This increase in LAG3 expression was not induced in conditioned media (Fig. 6E), indicating a cell–cell interaction was needed to induce receptor expression.

Upregulation of PD1 and TIM3 on CD3$^+$ T cells was also induced by HD MAMs compared to control and LD MAMs, although the difference was not statistically significant in the latter. CD3$^+$ T cell proliferation was significantly increased by co-culture with HD MAMs compared to control (Fig. 6E). In contrast, LAG3, PD1, and TIM3 expression was not induced in HD MAM conditioned media (Fig. 6F), indicating a cell–cell

**Fig. 4 | Tumor ECM alters the macrophage transcriptome. A** Schematic of macrophage decellularized tissue culture. Monocytes/macrophages from four separate blood donors were cultured for 14 days on high disease (MAM$^{HD}$) ($N=4$) or low disease (MAM$^{LD}$) ($N=4$) decellularized tissues. Total $N=31$, 4 blood donors, x4 high disease samples, x4 low disease samples (minus G198_D3 (Supplementary Fig. 19B)). Each sample represents six (wells) pooled culture samples from a 96-well plate, each well containing 200,000 cells. **B** Unsupervised cluster dendrogram using DE genes from HD or LD MAMs. **C–E** Bar plots of CIBERSORTx analysis using DE genes between HD and LD MAMs for **C** M0 macrophage, **D** M1 macrophage, and **E** M2 macrophage signatures. Data are presented as mean values +/− SD. Two-tailed unpaired t test. ****$p=0.000034$; M1 *$p=0.0368$; M2 *$p=0.0216$. $N=16$ HD MAM samples, $N=15$ LD MAM samples. **F** Flow cytometry histograms of transmembrane receptors expressed on HD and LD ECM cultured macrophages. Representative contour plot of HD and LD MAMs. **G** Bar plots of flow cytometry expression patterns of CD163 and CD209 between HD and LD MAMs, shown as percentage from the CD45$^+$ CD14$^+$ macrophage population. For example, CD163$^{low}$ and CD209$^{high}$

populations were selected from the bottom-right gating (M1-like) and CD163$^{high}$ and CD209$^{low}$ populations were selected from the top-left gating (M2-like) from Fig. 4F contour flow cytometry plots. Data are presented as mean values +/− SD. Two-tailed Mann-Whitney U test, $p=0.004$ and two-tailed unpaired t test, $p=0.0016$, respectively. $N=3$ LD MAM or HD MAM samples, with 3 technical repeats for each sample. **H** Heatmap of row z-scores of log2TPM gene expression for top 30 (right panel) up- and downregulated genes; total (left panel) up- and downregulated genes (adj. $p < 0.05$, logFC > |1|, protein coding). Samples split by ECM type (LD (blue) or HD (brown)) and donor 1–4 i.e., blood donors. **I** Bar plot of selected DE genes (from the list of top 30) between HD versus LD cultured macrophages. **J** LEGENDPLEX™ assay of secreted CXCL5. Each dot represents the mean value of sample duplicates for each blood donor per tissue. Data are presented as mean values +/− SD. Two-tailed Mann-Whitney U test, $p < 0.0001$. $N=16$ each. **K** LEGENDPLEX™ assay of secreted CXCL1 and CCL2 expression levels. Data are presented as mean values +/− SD. Two-tailed unpaired t test, *$p=0.0151$. $N=16$ each.

---

interaction between HD MAM and CD3$^+$ T cells was needed to induce receptor expression. In addition, when using MAM-conditioned media, no difference in CD3$^+$ T cell proliferation, or activation markers was observed (Fig. 6F). Taken together, in vitro co-culture assays support the transcriptomic analysis of HD MAMs and support a role in regulating infiltrating T cells within the TME.

## Discussion

The expanding repertoire of biological cancer therapies in recent years has highlighted the influence of the tumor microenvironment in host and therapy response particularly in carcinomas, and significant evidence has linked the composition of the tumor ECM with anti-tumor immunity and immunotherapy responses. There is a growing awareness that the ECM may have a direct role in tumor progression and host immunity, and therefore we set out to investigate whether the ECM directly educates the immune infiltrate. In this study, we show the tumor ECM composition in ovarian metastases and 15 other solid cancers associate with an M0 macrophage phenotype. This ECM composition is largely contributed from fibroblasts, however, tumor cells also produce these molecules, in some cases to similar levels seen in fibroblasts. The 5 members of the M0-ECM signature (Fig. 1), FN1, VCAN, COL11A1, SFRP2, and MXRA5, have been associated with cancer progression, through markers of prognosis[41,42] or the process of metastasis[43,44]. FN1, VCAN, and COL11A1 were also members of a signature we identified that associated with poor prognosis across many epithelial cancers. VCAN is highly involved in immune cell trafficking, and when in complex with hyaluronan, T cells move along the resulting fibers[45,46]. COL11A1 is associated with poor prognosis in several cancer types, and may have potential as a prognostic biomarker in pancreatic cancer[47]. Its function within the tumor microenvironment is not well investigated, but may be connected with activation of fibroblasts into a cancer-associated fibroblast (CAF) phenotype[48], and some reports indicate COL11A1 expression to be a CAF-specific feature[49]. Proteomic analysis on our library of ovarian metastatic tissues revealed five ECM composition groups (ECGs) which clustered based on the relative expression of ECM proteins, rather than the presence or absence of specific proteins. Therefore, it is the pattern of ECM that separates these groups. In the low disease tissues, we found two ECGs (ECG1-2), and we were surprised to see COL11A1 upregulated in ECG1, a group of tissues with low disease which at least from histopathological analysis appear normal and similar to the other low disease group ECG2. This may indicate this group of tissues are already undergoing ECM remodeling predisposing them for tumor colonization, or may be due to a separate parameter such as patient age. These concepts were recently investigated using bulk and single-cell transcriptomics datasets of squamous cell carcinomas[50] where ECM changes were found to be predictive of premalignant progression, of which COL11A1 was one of several markers identified. Comparing the age of patients, ECG1

tissues had a trend to be from younger individuals. Further investigations to explore early changes in tumor matrix or changes with age would benefit from the analysis of true healthy control tissue from cancer-free patients. High disease tissues were separated into three ECGs (ECG3-5) based on the pattern of ECM detected, which correlated with immune infiltrate, and in particular a high macrophage presence in tissues composed of high M0-ECM. These composition signatures could be reflected in the circulating fragments of ECM detectable in blood or urine and may have utility in cancer prognostics or diagnostics. There is significant interest in using ECM for prognostics and diagnostics, particularly within the collagen family of molecules[47,51–55].

To test whether the ECM was supporting the generation of the M0 TAM population we developed a decellularized model of ovarian cancer omental metastatic tissue. Inspired from previous work decellularizing ovarian tissues in mice[56], we optimized the methodology to decellularize human omental tissue while maintaining the native ECM composition and spatial arrangement of fibers. We characterized the ECM of the ovarian metastatic tissues using an ECM-focused proteomics approach[36]. Other approaches are available, and are used depending partly on the type of tissue being analyzed. A comparison of decellularization and ECM extraction techniques for proteomics analysis was recently performed[57], providing a useful starting point for future ECM-focused omics analysis. In addition, the cocktail of enzymes used could also prove important to optimize the coverage of peptides detected within a protein.

Using a decellularized tissue model of ovarian omental metastases, where the composition and structure of the tumor ECM is maintained, we were able to show that the ECM can directly educate macrophages that overlap with the transcriptomic profile of a prominent TAM population found in HGSOC. We suggest that the reason these TAMs associate with tumor ECM is because they are generated through an interaction of infiltrating monocytes with the tumor ECM, that we termed extracellular matrix-educated macrophages (MAMs). Of note, we found that decellularized tissue alone is sufficient to mature monocytes into macrophages, an observation that is supported by a study of monocytes cultured on a decellularized tissue model of colorectal cancer[58]. Macrophages educated on decellularized tissue undergo a phenotypic transition, shown by different patterns of marker expression and transcriptomic profiles, that is dependent on ECM composition. We show macrophages educated by tumor ECM (HD MAMs) have a TAM phenotype with significantly altered cell surface expression similar to alternatively activated macrophages and a transcriptomic signature comprised of immunoregulatory and tissue remodeling effectors including ARG1, CD36, MSR1, and MARCO. Analysis using a spectrum model of macrophage polarization revealed that neither MAM phenotype most strongly associated with stimuli linked to M1 or M2 polarization, which was in line with the analysis on whole cancer tissues (Fig. 1). These data suggest that HD and LD MAMs

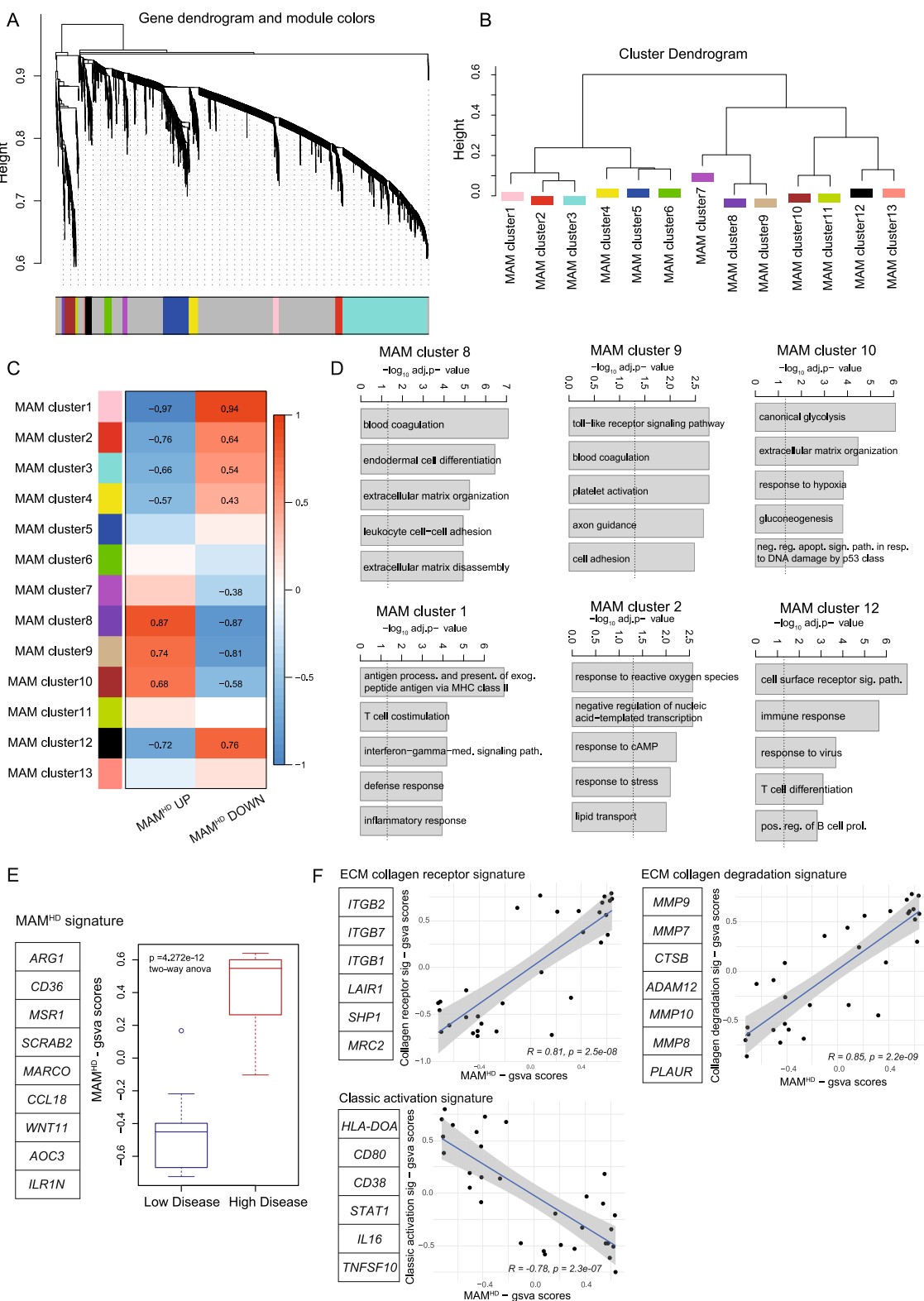

undergo transcriptional reprogramming, however, overall these changes are not consistent with a strict M1-M2-axis re-polarization. In line with these data, there is increasing evidence in the literature that while most TAMs are polarized towards an M2-like phenotype[22,23], they also exist as a spectrum of subtypes influenced by their interactions within the TME[15,16,37] HD MAMs have an immunoregulatory effect on T cells in their ability to alter the expression of activation/exhaustion markers on CD3[+] T cells and CD3[+] T cell proliferation. In co-culture

studies, we found HD MAMs upregulated classical markers of activation on CD3[+] T cells via direct cell–cell interaction. In the present study, the mechanism of how ECM educates the observed macrophage phenotype seen has not been investigated, however, transcriptomic data points to two potential mechanisms, both receptor-ECM interactions: there is leukocyte-associated Ig-like receptor 1 (LAIR1) which may signal immunosuppressive gene programs when activated through collagen binding[59], or sialoglycan-siglec signaling[60] where

**Fig. 5 | HD MAMs infer immunoregulatory phenotype. A** WGCNA of human macrophages cultured on HD and LD decellularized tissue showing clusters of co-expressed genes as dendrogram. Numbers indicate different modules (gene programs). *N* = 16 HD MAM samples, *N* = 15 LD MAM samples. **B** Cluster dendrogram of module eigenvalues (MEs) showing associated gene programs from WGCNA of human macrophages cultured on high and low disease decellularized tissue. *N* = 16 HD MAM samples, *N* = 15 LD MAM samples. **C** Heatmap of association of ME with gsva scores of the HD vs. LD, up and down. Pearson's r-values with two-sided alternative hypothesis testing are noted for the significant correlations *p* < 0.05. *N* = 16 HD MAM samples, *N* = 15 LD MAM samples. **D** Bar plots for significantly enriched gene ontology biological processes in clusters 8, 9, 10, and 1, 2, 12 gene programs. Broken line denotes adjusted *p* = 0.05. Hypergeometric test with multiple comparisons correction using the Bonferroni method. *N* = 16 HD MAM samples, *N* = 15 LD MAM samples. **E** Boxplot of selected DE HD MAM genes termed HD MAM signature. Boxplots illustrate median (center of the box) with the upper (Q3: 75th percentile) and lower (Q1: 25th percentile) quartiles (ends of the box); The whiskers correspond to Q3 + 1.5 x Interquartile Range (IQR) to Q1 − 1.5 x IQR; Dots beyond the whiskers show potential outliers. Two-way ANOVA, *p* = 4.272e−12. *N* = 16 HD MAM samples, *N* = 15 LD MAM samples. **F** Correlation scatter plots of gsva scores from gene lists. GSVA scores were calculated from the MAM RNAseq dataset using the gene lists indicated next to the scatterplots and R package gsva. Correlation coefficient corresponds to Pearson's correlation. Line was fitted with linear model (lm) with 95% confidence interval displayed. *p* values and correlation coefficients correspond to Pearson's method, with two-sided alternative hypothesis testing. *N* = 16 HD MAM samples, *N* = 15 LD MAM samples.

sialoglycans upregulated within tumor tissue may be ligands for immune-inhibitory siglecs on infiltrating macrophages. ECM education of some populations of TAMs is supported by other work which has shown tumor-associated glycoproteins like tenascin-C[61] and the mucin glycoform MUC1-ST[60] can induce TAM-like phenotypes and inhibit the function of other immune cells. Tissue stiffness has also been implicated in driving macrophage phenotype and based on our previous studies ovarian metastatic tumor tissue is stiffer (10-20kPa) than low disease tissue (0.1–1 kPa). The data presented here are supportive of the hypothesis that tumor ECM has a direct effect on tumor immunity. In the present study, we have concentrated on macrophages, however, there are other immune cell – ECM composition associations within the analysis presented in Fig. 1. Whether the ECM is also capable of driving certain phenotypes in these immune cells has not been tested, however, we suggest the decellularized tissue model would be suitable to test hypotheses involving other immune cells. The decellularized model allows the cellular component of the TME, which has been documented to stimulate macrophages, to be removed from the ECM in order to study direct ECM effects on macrophages, however, it may also be suitable for multi-cell type studies as well as allowing the cellular aspects of the TME to be recapitulated in as little or as much detail as required. As more attention continues to build on therapeutic targeting of the tumor microenvironment, we suggest the decellularized tissue model presented here is a useful research tool to help understand the challenges presented by the TME and may help support the development of the next cancer therapies.

## Methods

### Computational integration of immune signatures with tissue composition

First, we conducted a Spearman correlative analysis between disease score defined as percentage of tissue area occupied by malignant cells and stroma, a previously reported tumor ECM signature termed as 'matrix index' and the CIBERSORTx or xCell immune signatures from the RNA-seq data from 32 HGSOC patient tissues[1]. This analysis identified M0 macrophages as significantly positively associated with extent of disease and tumor matrisome. We next looked to see if we could elucidate specific interactions between immune populations and individual ECM molecules. To do this, we utilized a proteomic pipeline categorizing all core ECM proteins and associated molecules to identify 145 matrisome molecules found in both transcriptomic and proteomic datasets[1]. We then performed a Spearman correlative analysis to integrate the CIBERSORTx and xCell immune signatures with patient matched matrisome transcriptomic and proteomic datasets to identify significant immune-gene and immune-protein associations. Five ECM molecules were found in this analysis to be significantly positively associated with disease score and the M0 macrophage signature using both CIBERSORTx and xCell approaches at both the transcriptomic and proteomic level (FN1, VCAN, MXRA5, COL11A1, SFRP2).

### Ethics approvals for tissue samples

Metastatic ovarian cancer patient samples were kindly donated by women undergoing surgery at Barts Health NHS trust between 2010 and 2017. All tissue obtained was deemed by a pathologist to be surplus to diagnostic and therapeutic requirements and was collected under the terms of Barts Tissue Bank (HTA license number 12199. REC no: 10/H0304/14). Each patient gave written informed consent. The work was conducted in accordance with the Declaration of Helsinki and International Ethical Guidelines for Biomedical Research Involving Human Subjects (CIOMS). Patient information is detailed in Supplementary Dataset 5.

### Blood samples

Use of anonymised blood samples obtained from Cambridge Biosciences was covered by ethical review board (IRB) approval and informed consent. Approved uses include the isolation of white blood cells for scientific and medical research, including genetic analysis, at a university or pharmaceutical company.

### Tissue decellularization

Decellularization involved a series of washes first using hypotonic buffer (10 mM Tris, 5 mM EDTA, 0.1 mM phenylmethylsulfonyl fluoride (PMSF) (Sigma-Aldrich, P7626-250MG), pH 8) for 4 h at room temperature (RT), 100% acetone (Sigma-Aldrich, 179124) 16–18 h at 4 °C, 2.5 mM or 4% Sodium deoxycholate (Sigma-Aldrich, D6750-100G) for 4 h at 4 °C, for LD or HD disease respectively, and nucleic degradation solution (50 mM Tris, 1 mM Magnesium chloride ($MgCl_2$) (Sigma-Aldrich, M8266-100G), 0.1% BSA and 40 units/mL Benzonase nuclease (Sigma-Aldrich, E1014-5KU), pH 8) for 20 h at 37 °C. PBS 1% P/S washes were used between each buffer and at 4 °C for 2 days prior to tissue use for in vitro cultures. DNeasy® Blood and Tissue kit (Qiagen, 69504) following manufacturer's instructions. Nucleic acid (DNA content) was quantified using a Nanodrop 2000 spectrophotometer (Thermofisher). H&E stained sections were quantified using QuPath tissue imaging software that used a cell detection system to determine total cell number.

### Tissue proteomics

The ECM component was enriched from frozen whole decellularized tissue sections (20 × 30 μm sections, approximately 40–50 mg of tissue) and analyzed as previously described[36] and included solubilizing decellularized tissues in 8 M urea and Lys-C protease aided-digestion. Briefly, the extracted proteins were reduced, alkylated, and digested with trypsin. Peptides were separated by nanoflow ultra-high pressure liquid chromatography (UPLC, NanoAcquity, Waters) and analyzed by mass spectrometry using a LTQ-Orbitrap XL mass spectrometer (Thermo Fisher Scientific). Proteomics raw reads were aligned using MASCOT database[62]. Proteomics data provided in Supplementary Dataset 6.

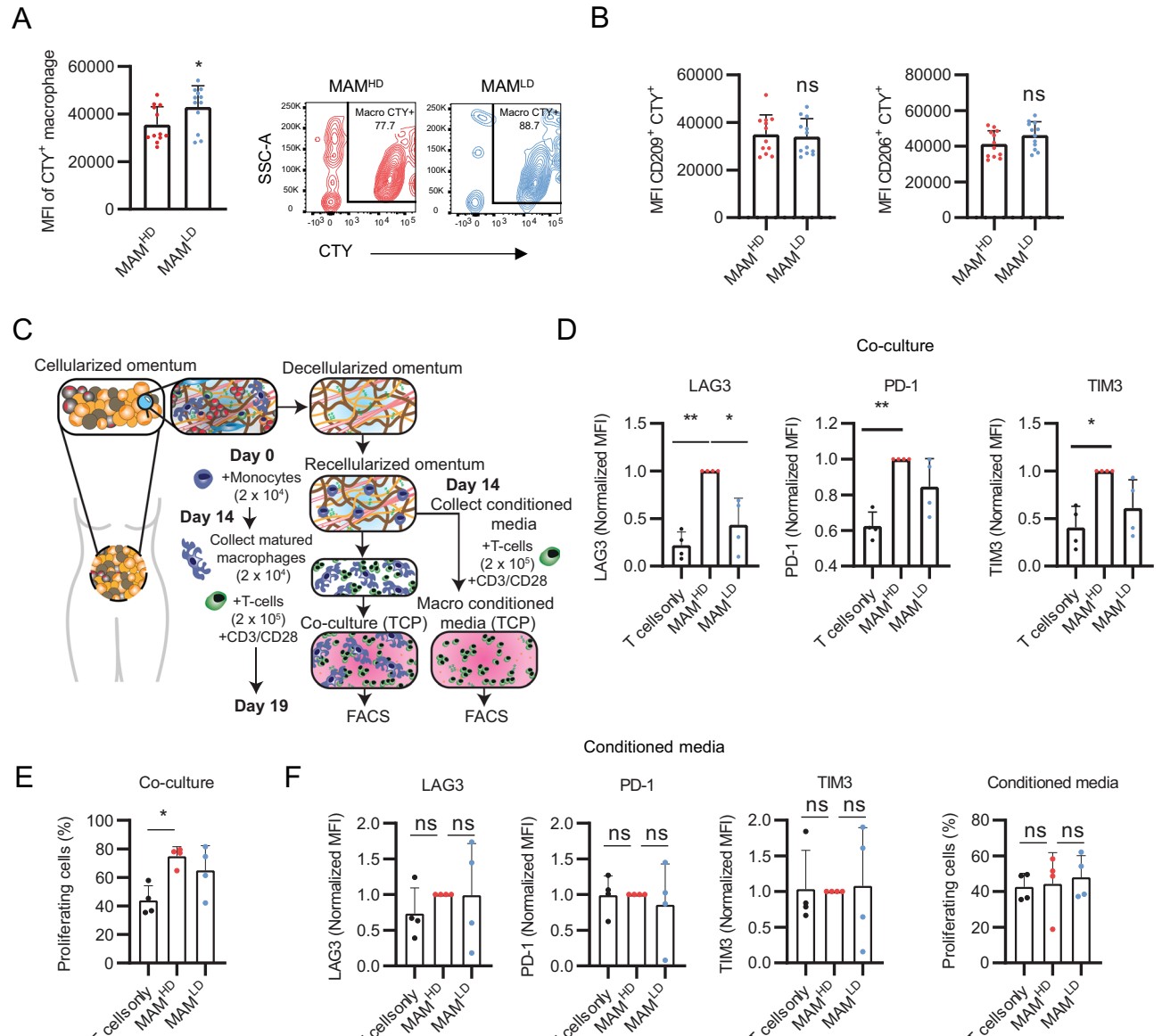

**Fig. 6 | HD MAMs have a reduced phagocytic response and alter T cell activation in the presence of CD3 and CD28 stimulation.** MAMs were cultured for 14 days and K562 cells were cultured for 5 days prior to use in phagocytosis assay. Cell types were mixed for phagocytosis assay. **A** Flow cytometry analysis of mean fluorescence intensity (MFI) of HD (MAM$^{HD}$) and LD MAMs (MAM$^{LD}$). Data are presented as mean values +/− SD. $p = 0.02$. Two-tailed Mann-Whitney U test used. Representative contour plots of CTY$^+$ macrophages cultured between HD and LD ECM. $N = 4$ LD MAM or HD MAM samples, with 3 technical repeats per sample. **B** MFI of CD209$^+$ and CD206$^+$ CTY$^+$ cells. Data are presented as mean values +/− SD. Two-tailed unpaired T-test used. $p = 0.77$, $p = 0.11$, respectively. $N = 4$ LD MAM or HD MAM samples, with 3 technical repeats per sample. **C** Schematic of MAMs and T cell co-culture workflow. Flow gating strategy provided in Supplementary Fig. 24. **D** Normalized MFI of LAG3, PD1 and TIM3 expression on CD3$^+$ T cells as assessed using flow cytometry after 5 days culture alone or co-culture with HD or LD MAMs. Data are presented as mean values +/− SD. One-way ANOVA followed by Dunnett's multiple comparisons test. LAG3 **$p = 0.0026$, *$p = 0.0444$, PD-1 **$p = 0.0039$, TIM3 *$p = 0.0209$, $N = 4$. **E** Percentage of proliferating CD3$^+$ T cells as assessed by Cell Trace Violet dilution using flow cytometry after 5 days culture alone or co-culture with high or low disease MAMs. Data are presented as mean values +/− SD. One-way ANOVA followed by Dunnett's multiple comparisons test. *$p = 0.013$, $N = 4$. **F** Normalized MFI of LAG3, PD-1, and TIM3 expression on CD3$^+$ T cells and percentage of proliferating CD3$^+$ T cells as assessed using flow cytometry after 5 days culture alone or co-culture with high or low disease MAMs conditioned media. Data are presented as mean values +/− SD. One-way ANOVA followed by Dunnett's multiple comparisons test. $N = 4$.

## Decellularized tissue macrophage culture

Sliced decellularized tissues were equilibrated with DMEM 1% P/S (no serum) at 4 °C overnight. Decellularized tissues were removed from DMEM 1% P/S and excess liquid removed. Decellularized tissue slices were placed into the wells of 96-well plate. To the decellularized tissues, 25 μL of isolated monocytes were added to the center of the tissue at seeding density of $2 \times 10^5/25$ μL. The monocytes were incubated with the decellularized tissue at 37 °C for 2 h to allow the cells to attach to the tissue. After 2 h, 200 μL DMEM supplemented with 10%

human AB serum (HS) (Sigma-Aldrich) and 1% P/S was added carefully to the cultures, not to disturb the tissue adhered cells. Cultures were maintained for 14 days.

## ECM IHC staining score analysis

For ECM tissue staining analysis, a staining score was developed to accurately assess ECM IHC staining between tissues. Definiens® defined staining intensities, namely marker area thresholds (μm²) between three categories, low, medium, and high. Low, medium, and high marker area

threshold categories were assigned as 1 (low), 2 (medium), or 3 (high). Marker area thresholds of positively stained sections were summed and multiplied by their respective scores. The sum was divided by total tissue area ($\mu m^2$) to account for the differences between tissue sizes. The staining score equation: $((SumLow \times 1)(+SumMedium \times 2) + (SumHigh \times 3))/(Tissue\ area\ sum) \times 100 = Staining\ score\ (AU)$.

## Quantifying ECM fiber diameter and alignment

Quantification of fiber diameter and alignment was performed on SEM micrographs using ImageJ version 1.50i (NIH). Three fields of view were chosen at random in cellularized and matched decellularized tissue micrographs ($N = 15$). Fiber diameter and fiber orientation angles were recoded (minimum 30 fibers quantified per field of view). The alignment index (AI) for each field of view was calculated as follows: where $N$ is the total number of fibers quantified per field of view, $\theta$ is the fiber angle (rad), $\theta_{avg}$ is the average orientation angle among the collagen fibers of the said field of view. When collagen fibers are randomly aligned AI will equal 0, whereas aligned fibers will have AI = 1.

## Masson's trichrome stain for collagen alignment

Masson's trichrome stain was performed and collagen ECM patterns were analyzed using The Workflow Of Matrix BioLogy Informatics (TWOMBLI) as described before (Wershof et al.)[32]. This analysis includes two tissue samples (G82, G278) not present in the original tissue library of 39 human omental ovarian metastatic tumor samples.

## Phagocytosis assay

K562 (ATCC CCL-243) cells, an immortalized myelogenous leukemia cell line which are lysed rapidly by Fc receptor-positive leukocytes (i.e., sensitive for killing assays) and do not phagocytose or mediate antibody-dependent phagocytosis[63–65], were grown for 5 days and collected from a T75 flask and then stained with CellTrace Yellow (CTY) in PBS, 1:10,000 for 20 min at 37 °C then 20 mL RPMI was added and incubated 5 min at 37 °C. CTY+ K562 cells were centrifuged and resuspended in RPMI at $5 \times 10^5$ cells/mL and plated onto an ultra-low attachment plate at 25,000 cells/well and incubated with anti-CD47 antibody (Cat: 16-0479-85, ThermoFisher) at 40 μg/mL for 1 h at 37 °C. In the meantime, MAMs were collected from decellularized tissues and seeded into a fresh ultra-low attachment plate at 10,000 cells/well. Live K562 cells/ well were washed in RPMI, resuspended in 100 μL RPMI, and combined with MAMs for 2 h at 37 °C. Cells were then washed and stained with a flow cytometry antibody cocktail and analyzed by flow cytometry. Staining panel used: CD206 (FITC, 1:100, Cat: 321104, Biolegend), CD209 (APC, 1:100, Cat: 330107, Biolegend), HLA-DR (AF700, 1:100, Cat: 307626, Biolegend), CD36 (BV421, 1:100, Cat: 336229, Biolegend), CD38 (BV605, 1:150, Cat: 356641, Biolegend), CD86 (BV650, 1:150, Cat: 305428, Biolegend), CD11b (AF594, 1:200, Cat: 301340, Biolegend), CD14 (PE-Cy5, 1:150, Cat: 15-0149-42, Invitrogen), CD204 (PE-Cy7, 1:100, Cat: 371907, Biolegend) and Zombie NIR (1:1000, Cat: 423106, Biolegend); phagocytic events were captured using CTY on the PE channel.

## T cell assay

Pan-T cell negative selection was performed by magnetic cell sorting (EasySep™ Human T Cell Isolation Kit from STEMCELL) from frozen PBMCs of autologous samples according to the manufacturer's instructions. CD3+ T cells were then resuspended at $2 \times 10^6$ cells/ml in PBS and stained with Cell Trace Violet (1:2000, Invitrogen) for 20 min at 37 °C. Cells were then washed twice in complete medium (RPMI supplemented with 10%FBS and 1% Penicillin-Streptomycin), seeded in a 96-well plate at $2 \times 10^5$ cells per well either alone or in presence of $20 \times 10^4$ HD or LD MAMs and were activated with 1 μg/ml anti-CD3 (Clone: OKT3, Cat: 317326, Biolegend) and 5 μg/ml anti-CD28 (Clone: CD28.2, Cat: 302943, Biolegend). At day five FACs analysis was performed to evaluate CD3+ T cell activation and proliferation.

## Laser capture microscopy

Frozen tissue sections from human omental metastasis (14 micron) were cut onto LCM membrane RNAse-free slides (previously activated under UV light for 30 min). Tissue sections were stained with hematoxylin (a few drops of hematoxylin was added to each section which was then immediately washed in DI water (a few dips), then tap water (a few dips), then submerged in 70% EtOH (30 sec), then 100% EtOH (1 min), and finally xylene (30 s) before air drying). Slides were kept on dry-ice where possible, and prior to laser capture. Tumor and stromal areas were then captured using a PALM laser capture microscope (examples of areas taken are shown in Fig. S5). Total cut time per slide was 30 min max. Cut pieces were pooled from 3–6 tissue sections, and total RNA isolated using Qiagen MinElute columns including on-column DNAse treatment as per the manufacturers protocol. Samples sent for sequencing had RNA concentrations in the region of 1800–18000 pg/μL and RIN numbers >7.

## Cell-derived matrices (CDMs)

In order to decellularize in vitro cultured cell lines, media from 14-day confluent cells (cultured with 50 μg/ml of L-ascorbic acid 2-phosphate (AA2P), dissolved in $dH_2O$)) was first removed and cells were washed using PBS. Cells were then incubated with 37 °C warmed extraction buffer (20 mM $NH_4OH$, 0.5% Triton X-100 in PBS) for 20 min or until cells showed signs of lysis (visualized by light microscope). Cells were next washed with PBS 70 several times to remove cell bodies. To degrade the cellular DNA, 5–8 mL of DNase I (Sigma-Aldrich) at 10 μg/mL in PBS was incubated with cells at 37 °C for 2 h. Following this, cells were washed with PBS 1% P/S several times. The resulting CDMs were stored at −80 °C.

## Statistical analysis

All statistical analyses were performed using either GraphPad Prism software version 8.3.0 for Windows, GraphPad Software, San Diego, California, USA, www.graphpad.com or the statistical programming language RStudio (2022.02.3 + 492 "Prairie Trillium" Release) and R (version 4.1.1) using the following software plugins: Hmisc for correlation analysis, gplots for correlation scatter plots, ggplot2 for bar charts, pheatmap for heatmaps, dendextend for dendrograms and ggpubr for editing figures to publication standard. Multivariate correlations were calculated using Spearman's or Pearson's correlation as appropriate, applied on linear or log transformed data, where $p < 0.05$ is considered significant unless otherwise specified and indicated with asterisk: $*p < 0.05$, $**p < 0.01$, $***p < 0.005$. Statistical tests used were indicated in the figure legends.

## Reporting summary

Further information on research design is available in the Nature Portfolio Reporting Summary linked to this article.

## Data availability

The RNAseq data have been deposited in NCBI's Gene Expression Omnibus[66] under the accession number GSE186145. The mass spectrometry proteomics data have been deposited to the ProteomeXchange Consortium via the PRIDE[67,68] partner repository under the dataset identifier PXD036940. Source data are provided with this paper.

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

## Acknowledgements

O.M.T.P. is a recipient of a Centre for Inflammation and Therapeutic Innovation (CiTI) funded studentship (for E.H.P.), a CRUK & Credit Suisse career establishment award (grant code: A27947, funding E.J.T., V.G., Y.L.), a *Barts Charity & Against Breast Cancer* studentship (MGU0499, for P.H.), and an MRC iCASE PhD studentship (for M.B.R.). M.P. is recipient of a grant of Foundation Institute of Pediatric Research Città della Speranza (grant number 27/01).

## Author contributions

E.H.P., E.J.T., A.W., and O.M.T.P. were involved in conception or design of the work; E.H.P., E.J.T., M.M., E.M., C.B., M.B.R., E.P., P.H., V.G., Y.L., G.M., V.R., P.C., C.T., M.P., M.L., J.R., H.L., A.W., and O.M.T.P. were involved in the acquisition, analysis, or interpretation of data; E.H.P., E.J.T., A.W., and O.M.T.P. drafted the work or substantively revised it. All authors read and agreed on the content of the paper.

## Competing interests

The authors declare no competing interests.
