## [Peer Review File · Nature Communications]

Extracellular matrix educates an immunoregulatory tumor macrophage phenotype found in ovarian cancer metastasisREVIEWER COMMENTS

Reviewer #1 (expertise in ECM proteomics):

The manuscript by Puttock et al. explores the relationship between various aspects of the tumor stroma in ovarian tumors. Overall, I found this manuscript to be well written and to follow a logical path of reasoning. The work is exciting in that it identifies a signature in the local tumor microenvironment and this signature is linked to a macrophage subpopulation and connections are made to patient outcomes. Some validation is provided in population survival data. A relatively novel aspect of this work includes the development and used of a decellularized tumor tissue to study the effects of ECM composition and architecture on cell phenotypes. Immune cell activation was one component explored and this shed light on a mechanism of tumor immune cell evasion. I recommend the manuscript for publication once the following items have been addressed. In the future, please include figure legends with figures (supplemental files). This may be a requirement of the journal.

Matrisome proteins are discussed and shown in several locations in the manuscript and supplemental files. Please always indicate if the data are derived from transcriptome or proteome (MS, IHC, etc) methods. One is left to assume in a few locations.

Line 143 – please mention the number of patients here.

Line 145 – “online resource” (ref 30)

Line 150 – The reference to further analysis across 19 carcinomas should be expanded on a little. I assume that it may be obvious if one is familiar with the online resource mentioned above?

Line 166-169 – levels measured by IHC?

Line 203 – How was it determined that 5 groups was the correct number?

I am not completely sold on the disease scoring system used. Is there an indication that the high score tumors give rise to more aggressive/advanced/poor outcomes – ref one is given? Please elaborate in the text.

Line 218. Which 4 proteins are the authors referring to? It was a 5 earlier in the manuscript unless it is the 4 that the authors used to evaluate the decellularization with (line166-169)

Line 226 – This statement doesn't appear to be consistent with earlier points in this paragraph or the data presented.

Line 238 – “tissue library” this is ambiguous. While it can be figured out what tissues the author is referring to I recommend a slightly more descriptive name.

Line 268 – ECGs1&2 are now being referred to as adjacent tissue? Please show that this tissue is adjacent to tumor and not a sub-category of tumor. Is the tumor margin included? How far from the tumor on average are these regions? Is this non-malignant?

Line 293 – Age matched control tissue was not available and tested?

Line 427 & Figure 1F. Please include the individual dataset results (even if in supp files). With the long half-life observed in many ECM proteins I would not be surprised if there wasn't concordance with transcriptomics.

Line 444-446. I believe the methods in ref 31 include adding urea to solubilize difficult ECM proteins and Lys-C protease is used to help in the digestion. Where these steps performed here? Please consider the limitation that a subset of ECM that is accessible using these methods was observed (ie characterization was not comprehensive).

Reviewer #2 (expertise in ovarian cancer RNA-seq):

The authors associated ECM composition in omental HGS-ovarian cancer tumors with deconvolution-predicted proportions of immune cells by using data from their earlier publication. They found an M0 or general macrophage population, based on only the deconvoluted expression data, to be associated with certain ECM proteins that they refer to as “ECM signature associated with M0 macrophages”. They show that the expression of these ECM proteins alone or combined are associated with poor prognosis in ovarian and many other epithelial cancers in TCGA data. The authors then set up a methodology to decellularize the samples and show that the decellularized samples retain ECM proteins and tissue fiber structures while losing nuclei and DNA content. The ECM proteins of interest are then shown to be present on ECM from samples that contain stroma (fibroblasts) and/or tumor cells but not in pure adipose samples, suggesting that they are not adipose derived.

The authors use the decellularized ECM models to study how it affects macrophage cellular states on M0-M1-M2 axis and find that ECM from tumor containing samples turns monocytes to only M0/M2 macrophages while adipose-derived ECM produces also M1 macrophages. Finally, the authors show that tumor-ECM educated macrophages induce repressive markers in T cells via direct cell-to-cell interactions. Adipose-ECM educated monocytes induce the repressive T cell markers at lower levels, while both induce T cell proliferation when in direct contact.

Overall, the papers first part (Fig 1) where the authors describe a connection between certain ECM and M0 macrophages seems weak as it relies too much on immune deconvolution analysis without even a basic marker-based sanity check on non-deconvoluted data to show whether M0 and indicated ECM markers co-occur in the studied or other omental HGSC cohort. The latter part with effects of decellularized models to macrophages, and these macrophages' effects to T cells is solid work and would be better off without the currently shaky basis that Fig. 1 forms.

Please find below more specific comments:

1. Description of the QC on deconvolution methods is insufficient and conclusions are not supported by the data:

"As anticipated, deconvolution of our bulk RNA-sequencing (RNAseq) dataset using analytical tools CIBERSORTx and xCell (which were designed to analyze RNAseq data) accurately estimated the abundance of CD8+ (R = 0.57 and 0.60, respectively), CD68+ (R = 0.36 and 0.46, respectively), CD3+, CD4+, and CD45RO+ immune cell counts."

Only the CD8 predictions are relatively robust for all methods whereas no method predicts CD4+ correctly, and even CD68+ that should be relevant for any macrophage correlation analysis is relatively weak, although significant. The statement that also CD3+, CD4+, and CD45RO+ immune cell counts were accurately estimated is not supported by the data. Suppl.Fig 1D and 1E are not referred and miss description in main text/methods to explain how the comparison was made.

2. Please more clearly state why "the composition of the tumor ECM associates most strongly with a M0 macrophage subtype". This is not evident from Figs 1B-F.

3. Please also directly assess expression of M0 markers in relation to ECM composition from transcriptomic (and if possible, also proteomic data), similar to seen in Fig 1D&E, especially as neither deconvolution method predicts the proportion of CD68+ cells particularly well.

4. The M0 macrophage signature should be assessed in GEO available HGSC scRNA-seq or spatial transcriptomics data; do the M0 macrophages overlap with M2s or M1s?

5. The term "poor prognostic M0 TAM signature" is misleading. The poor prognosis association is shown for ECM sub-signature (that associates with M0 in deconvoluted data) but not for M0 TAMs.

6. Higher percentage of PAX8+ cells and stroma in a tissue sample does not equal to disease progression; please rephrase: " There was a significant difference in the disease scores between ECG1-2 when compared against ECG3, 4, or 5, indicating the change in ECM composition that occurs as disease progresses." It appears that ECG1 & 2 simply define normal adipose tissue (and are later referred to as "adjacent tissue"); please refer to it as such or explain in the manuscript why this is not the case.

7. Figure 3 does not support the association between M0 macrophages and the ECM molecules of interest. In Fig 3E and 3F, all sample classes have macrophages as the most prominent class, and macrophage total count or proportion does not seem to be reflected in the strength of "M0 associated macrophage signature" proteins.

8. Supplemental Fig 19 and Fig 5 are too complicated for the reader with the color codes for WGCNA modules. Please name the clusters by functional annotations instead to help the reader.

9. It is unclear from which data the gsva scores in Fig 5F are calculated; please add description.

10. The claim of tissue remodelling capabilities in M0 is supported by the data shown; modify statement to make claims of only tissue remodelling-related gene expression as the data supports that.

11. Fig 6 sub-panels and legend seem to be scrambled.

Reviewer #3 (expertise in ECM and decellularised tissue models):

Overall, this manuscript by E. H. Puttock et al., describing a role for extracellular matrix proteins in the induction of immune suppression in human ovarian cancer, is novel, rigorous, and substantially advances our understanding of the role of the ECM in tumor cell immune surveillance. The core of the study is the prospective collection of fresh, patient derived ovarian cancer tissue specimens (HGSOC) and their analysis by gene expression and ECM proteomics. Using a multitude of RNA sequencing deconvolution and pathway analysis tools, associations between immune cell populations and ECM composition are elucidated. Further, how these associations change between adjacent normal, non-cancer pathologies and cancer are also delineated. Based on these omic-level studies, specific ECM compositions that correlated with tumor associated macrophages (TAMs) were identified. To demonstrate causality, fresh patient derived tissues were decellularized, and the resulting ECM scaffold used as a substratum to culture monocytes. In vitro, tumor derived matrices induced a macrophage phenotype similar to the TAM phenotype observed in patient samples. The authors refer to these ECM or matrix-educated macrophages as MAMs, and the determination that macrophages can be educated by human relevant, ovarian cancer ECM matrices is the major advance of this study. In an ex-vivo T cell activation assay (apparently antigen independent T cell activation, or at least in the absence of a model antigen), tumor educated MAMs induced T cell proliferation, activation and exhaustion markers more robustly than monocytes educated by decellularized adjacent normal tissues.

The paper has several strengths including the large number of patient derived tissues analyzed, and the inclusion of adjacent normal tissues and ovarian tissues with non-cancer pathologies. Another strength is the thoroughness by which the authors confirmed decellularization of the matrix scaffolds and demonstrate the integrity of these scaffolds after decellularization. Finally, a strength is the breadth of analyses employed, a requisite to undertake analysis of the tumor microenvironment in clinically relevant tissues.

The manuscript also has some significant weaknesses, all of which, in this reviewer's opinion, can be remedied by revision of the current document. The primary weakness is the lack of clarity with respect to experimental design, making data interpretation difficult. The results, methods, and figure legend sections provide insufficient amount of information to interpret the data. The results section is exceptionally sparse on experimental design details. Another weakness is the sparse discussion section, which mostly reiterates the results section. Key observations are not sufficiently discussed in light of the literature. Why might expression of their "core" ECM molecules FN1, COL11A1, VCAN, MXRA5, SFRP2, COL1A1, CTSB, and CS, be upregulated in ovarian cancer, and influence macrophage phenotypes? How might these molecules or the pathways they elicit be targeted, as the authors suggest? What are the limitations of their study? Due to low solubility, many ECM proteins are underrepresented in proteomic databases, and this is particularly true for the fibrillar collagens. A stepwise extraction with CHAPS and high salt, guanidine hydrochloride and chemical digestion with hydroxylamine hydrochloride (HA) has been shown to be optimal for ECM proteomics (see work of Kirk Hansen, for example <https://doi.org/10.1016/j.mcpro.2021.100079>). It appears from the methods section that this optimized approach was not used. Given the importance of the ECM proteomics data to the overall interpretation of the current study, additional information on sample preparation and inherent limitations to the ECM proteomics data need to be included in the methods and discussion sections, respectively.

Referencing past studies, especially related to methods, are sparse.

Pioneering work by others in the field of decellularized matrices are not referenced. Is the decellularization protocol used by the authors not based or inspired by any prior studies?

The rationale and details of the phagocytosis assay are sparse, and again references to others performing these assays are lacking.

The methods for scanning electron microscopy and matrix fiber characterizations are missing.

T cell activation assay also is not referenced. What T cell population is isolated with this kit, CD3+, CD4+ or CD8+ T cells?

Specific comments related to data presentation are as follows:

Figure 1

Are Figures 1B & C combining proteomics ECM composition with immune gene signatures? Or are these analyses ECM gene expression to immune gene expression associations?

Is the primary goal to show correlations between ECM and immune signatures, or to show that these correlations change with disease state? It seems like the data presented in this figure are 'richer' than described. In Fig 1B & C, can the data be color coded by case/stage of disease?

Rotating these figures to horizontal might help highlight the main point, which appears to be differential ECM proteins signatures between immune cell types? Are the authors suggesting that each immune cell subtype has its own ECM microenvironment? Or that different disease states have a unique ECM milieu, which associate with changes to the immune milieu? Please clarify.

Fig 1E - are any of the immune cell separations observed within these two distinct clusters statistically different, or are the meaningful differences observed between the two main clusters?

Fig 1G- based on proteomic analysis, 5 ECM proteins (FN1, COL11A1, VCAN, MXRA5, SFRP2) are used to delineate tumor high and low expression groups. Step 2 of the "cancer immunity cycle" is increased in high expressers. Do the authors know if all 5 ECM proteins are contributing to this difference or is one ECM protein dominating?

Figure 2

Three of the 5 ECM molecules of focus change between Figure 1 (FN1, COL11A1, VCAN, MXRA5, SFRP2) and Figure 2D (COL1A1, FN1, CTSB, VCAN and CS) without a clear explanation for why.

Figure 2F & G - methods for quantifying fiber diameter and alignment are missing.

Figure 2F- presumably the red cells are dying? Please clarify in the figure legend. It is unclear to this reviewer whether these data are simply artifacts of the cell culture system or of use to the interpretation of the in vivo biology they are trying to dissect.

Figure 3

It is unclear what is meant by disease score (disease profiling)-are these clinical parameters of disease such as tumor size, stage or a molecular signature? What does synergy mean in the figure title?

Are the data in Fig 3B defining ECG1-5, or have these categories been previously defined? The figure legend in Fig 3B reads as if these clusters were known previously, but the data appear to be used to define these 5 groups. Please clarify.

Is the M0 macrophage-associated ECM signature based on gene expression? The Y axis reads as if only FN1, COL11A1, VCAN, and MXRA5 protein levels are being quantified, and not an M0 gene signature. Please clarify.

How were immune cells/tissue in Fig 3E assessed, flow? IHC? State in figure legend or results text.

Figure 4

Figure 4B- in the legend, please state that the clustering is of monocytes after 14 days in culture on tumor and adjacent decellularized donor tissue and provide numbers of cases per condition.

Were any replicates done to confirm low intra-assay variability?

Figure 4F-is this flow data? The legend says histograms of receptor expression.

Figure 4G- presumably the low/low and high/high populations are either rare or did not change between tumor and adjacent decellularized matrices?

Figure 4H has two panels, and the distinction is not discussed in the text or legend. It is unclear what "donor" refers to in this figure.

Figure 4I-are these top 30 genes found to be enriched in MAMS, meaning expressed preferentially in monocytes cultured onto tumor decellularized stroma? Please clarify.

Figure 5

I don't find the color coding of data to be highly informative-I think Figure 5B and C do not add much value. It is unclear why some gene clusters (colors) are described more fully than others.

For example, why is the gene set defined as turquoise not discussed?

Figure 5E seems largely unrelated to the WGCNA data analyses, but important none-the less.

Please provide more details on how this select group of MAM signature genes were collated.

Figure 5F- it is unclear what is being correlated in these graphs-gene expressing to gene expression scores obtained fromwhat samples?

Figure 6

The overall design of this experiment is vague. Are macrophages engulfing CTY labeled tumor cells

in an ex vivo assay? Are live or dead tumor cells added? Are these data consistent with data shown in figure 1G, where antigen presentation is reported to be increased?
Figure 6D, E & F-please label control conditions on X-axis. Please clarify in results text or figure legend that data shown in D, E and F were obtained by flow cytometry.

Reviewer #4 (expertise in ECM biology and ECM in cancer):

This manuscript examines the matrixome in high grade serous ovarian cancer (HGSOC) omental metastases and also uses a decellularized/re-cellularized tissues to probe the ability of the matrix to "educated" tumor associated macrophages (TAMs). Overall, the work is thorough, performed rigorously, and presents convincing results to largely support the authors conclusions. However, I have a few comments that warrant some attention.

1) I have several questions regarding the fibers in the SEMs. The identities are unclear as in some of the panels they appear to be fibrillar collagen and in others more like basal lamina (Fig 2e). Yet the fibrillar orientation analysis appears to be the same. I recommend some immunostaining to identify these and clarify the analysis. Similarly, is there sufficient data to perform quantitative analysis of fiber? Lastly, as there have been several reports using optical microscopy of collagen alignment in the fallopian tubes, ovarian cortex and omentum, is the alignment data similar to those results?

2) I would like to see more on the criterion for identifying regions that are tumor and tumor adjacent (as used in several figures). While a mainly cellular criterion was used, it has been shown in the FT and primary ovary that collagen can be altered in regions of low cellularity, where the fiber morphology is highly distinct from normal tissues or distant normal regions in diseased tissues. Thus, more rigorously establishing this classification is important as it was used in a few contexts.

3) The large results showing differences in Col11 in low and high disease in the hazard analysis (Supp. Fig 7b) are interesting. There are several reports in the literature regarding Col 11 expression and remodeling, and it would be ideal to put the current findings in that context.

RESPONSE TO REVIEWERS' COMMENTS

Reviewer #1 (expertise in ECM proteomics):

Point 1. The manuscript by Puttock et al. explores the relationship between various aspects of the tumor stroma in ovarian tumors. Overall, I found this manuscript to be well written and to follow a logical path of reasoning. The work is exciting in that it identifies a signature in the local tumor microenvironment and this signature is linked to a macrophage subpopulation and connections are made to patient outcomes. Some validation is provided in population survival data. A relatively novel aspect of this work includes the development and used of a decellularized tumor tissue to study the effects of ECM composition and architecture on cell phenotypes. Immune cell activation was one component explored and this shed light on a mechanism of tumor immune cell evasion. I recommend the manuscript for publication once the following items have been addressed.

Response. Thank you for the positive review of our work, and the critical appraisal. We have addressed your points below.

Point 2. In the future, please include figure legends with figures (supplemental files). This may be a requirement of the journal.

Response. We have followed the guidelines from Nature Communications: “Text for figure legends should be provided in numerical order after the references.”

Point 3. Matrisome proteins are discussed and shown in several locations in the manuscript and supplemental files. Please always indicate if the data are derived from transcriptome or proteome (MS, IHC, etc) methods. One is left to assume in a few locations.

Response. We have now clarified where the data are derived in the text and figures throughout.

Point 4. Line 143 – please mention the number of patients here.

Response. We have clarified this now under Line 151 - “Laser capture microscopy of tumor and stroma areas from HGSOc tissues from **two patients in the same cohort** as the above analysis”

Point 5. Line 145 – “online resource” (ref 30)

Response. We have clarified this in the text on line 154 to now read “an online resource¹ (<https://kmplot.com/analysis/>)”

Point 6. Line 150 – The reference to further analysis across 19 carcinomas should be expanded on a little. I assume that it may be obvious if one is familiar with the online resource mentioned above?

Response. We have expanded to make clearer as suggested now on Line 159 - “Further analysis **using the KM Plotter for pan-cancer** across 19 carcinomas found 12 cancers (including ovarian cancer)”

Point 7. Line 166-169 – levels measured by IHC?

Response. Yes, and we have clarified now under Line 180-184 – “We next used IHC to assess the content of four ECM molecules (collagen 1 (COL1A1), FN1, VCAN, and cathepsin-B (CTSB)) and one glycosaminoglycan, chondroitin sulfate (CS), which were selected based on their high level of expression in cancer tissues² and/or their association with the macrophage phenotype described above (Figure 1).”

Point 8. Line 203 – How was it determined that 5 groups was the correct number?

Response. 5 groups was determined from the cluster analysis performed. We have expanded the text to make this clear on Lines 228-229 - “Hierarchical clustering separated the samples by their proteomic ECM composition which identified five ECM composition groups (ECGs) (Figure 3B).”

Point 9. I am not completely sold on the disease scoring system used. Is there an indication that the high score tumors give rise to more aggressive/advanced/poor outcomes – ref one is given? Please elaborate in the text.

Response. The disease score (DS) is a digital histopathology method that we first described in *Cancer Disc*, 2018². DS measures extent of disease present within a sample based on the area of tumor (PAX8 positivity) and stroma (hematoxylin staining of areas where there is stromal remodelling or existing stroma). Previously we found DS associated with an increase in immune infiltration, tissue stiffness, and ECM remodelling. It does not associate with clinical parameters such as grade or prognosis. We have clarified this in the main text:

Line 230 – “The disease score is a digital histopathology method we described previously², that measures extent of disease present within a sample based on the area of tumor (PAX8 positivity) and stroma (hematoxylin staining of areas where there is stromal remodeling or existing stroma). Previously we found disease score to be associated with an increase in immune infiltration, tissue stiffness, and ECM remodeling.”

Point 10. Line 218. Which 4 proteins are the authors referring to? It was a 5 earlier in the manuscript unless it is the 4 that the authors used to evaluate the decellularization with (line166-169)

Response. We have clarified this on Line 249 - “Individually, the ECM signature molecules tended to be highly expressed in ECG3 and ECG5 and appeared highest in ECG5, with the exception of COL11A1 which did not correlate with a specific ECG group(s) and SFRP2 which was not detected in this proteomic dataset”

Point 11. Line 226 – This statement doesn’t appear to be consistent with earlier points in this paragraph or the data presented.

Response. We have rephrased this statement, which now reads:

Line 257 - “Taken together, tumors with similar disease scores can have different ECM compositions.”

Point 12. Line 238 – “tissue library” this is ambiguous. While it can be figured out what tissues the author is referring to I recommend a slightly more descriptive name.

Response. We have clarified on Line 220 - “library of **ovarian cancer metastatic** tissues”

Point 13. Line 268 – ECGs1&2 are now being referred to as adjacent tissue? Please show that this tissue is adjacent to tumor and not a sub-category of tumor. Is the tumor margin included? How far from the tumor on average are these regions? Is this non-malignant?

Response. We agree adjacent is not the correct description of what these samples are, which is omental samples with a low level of disease as detected by histopathological analysis. The tissue samples of ECG1-2 were composed almost entirely of adipose tissue (Supplemental Figure 8D) and had ECM compositions comparable to normal tissues³ and low immune cell abundances (Figure 3E-F). To make this clearer in the text and figures we have changed where we have used tumor and adjacent to ‘high disease’ and ‘low disease’ respectively. Similarly, MAM^{Tumor} has been replaced with MAM^{HD} (high disease extracellular matrix-associated macrophages) and MAM^{Adjacent} has been replaced with MAM^{LD} (low disease extracellular matrix-associated macrophages) throughout the text and figures.

Point 14. Line 293 – Age matched control tissue was not available and tested?

Response. We did not specifically account for tissue age, and so the samples we have don’t allow us to test whether age contributes to the ECM composition of the tissues very well. We do agree that age could be an interesting parameter to consider, particularly in light of emerging work looking at post-translational glycosylation changes with age (glycan-age), and beautiful work recently published describing ECM profiles associated with cancer risk and prognosis that identified age related changes in ECM and cancer risk⁴. We have added a note to include this work in the discussion: Line 417 - **We were surprised to see COL11A1 upregulated in ECG 1, a group of tissues with low disease and at least from histopathological analysis appear normal and similar to the other low disease group ECG2. This may indicate this group of tissues are already undergoing extracellular matrix remodeling predisposing them for tumor colonization, or may be due to a separate parameter such as patient age. These concepts were recently investigated using bulk and single cell transcriptomics datasets of squamous cell carcinomas⁴ where ECM changes were found to be predictive of premalignant progression, of which COL11A1 was one of several markers identified. Comparing the age of patients, ECG1 tissues had a trend to be from younger individuals.**

Point 15. Line 427 & Figure 1F. Please include the individual dataset results (even if in supp files). With the long half-life observed in many ECM proteins I would not be surprised if there wasn’t concordance with transcriptomics.

Response. We have added the individual dataset results for CIBERSORTx in ‘Supplemental Table 2.’ And xCell in ‘Supplemental Table 3.’

Point 16. Line 444-446. I believe the methods in ref 31 include adding urea to solubilize difficult ECM proteins and Lys-C protease is used to help in the digestion. Where these steps performed here? Please consider the limitation that a subset of ECM that is accessible using these methods was observed (ie characterization was not comprehensive).

Response yes we have used the method that includes the addition of Lys-C protease. We have added a note regarding this in the methodology and have included more recent methodology for measuring ECM proteomics in the discussion.

Line 510 – described³⁵ and included solubilizing decellularized tissues in 8M urea and lys-C protease aided-digestion.

Line 429 - We characterized the ECM of the ovarian metastatic tissues using an ECM-focused proteomics approach³⁷. Other approaches are available, and are used depending partly on the type of tissue being analysed. A comparison of decellularisation and ECM extraction techniques for proteomics analysis were recently compared⁵⁴, providing a useful starting point for future ECM-focused omics analysis. In addition, the cocktail of enzymes used could also prove important to optimize the coverage of peptides detected within a protein.

Reviewer #2 (expertise in ovarian cancer RNA-seq):

Point 1. The authors associated ECM composition in omental HGS-ovarian cancer tumors with deconvolution-predicted proportions of immune cells by using data from their earlier publication. They found an M0 or general macrophage population, based on only the deconvoluted expression data, to be associated with certain ECM proteins that they refer to as “ECM signature associated with M0 macrophages”. They show that the expression of these ECM proteins alone or combined are associated with poor prognosis in ovarian and many other epithelial cancers in TCGA data. The authors then set up a methodology to decellularize the samples and show that the decellularized samples retain ECM proteins and tissue fiber structures while losing nuclei and DNA content. The ECM proteins of interest are then shown to be present on ECM from samples that contain stroma (fibroblasts) and/or tumor cells but not in pure adipose samples, suggesting that they are not adipose derived.

The authors use the decellularized ECM models to study how it affects macrophage cellular states on M0-M1-M2 axis and find that ECM from tumor containing samples turns monocytes to only M0/M2 macrophages while adipose-derived ECM produces also M1 macrophages. Finally, the authors show that tumor-ECM educated macrophages induce repressive markers in T cells via direct cell-to-cell interactions. Adipose-ECM educated monocytes induce the repressive T cell markers at lower levels, while both induce T cell proliferation when in direct contact.

Response. Thank you for the critical appraisal of the work. Below we have addressed the comments and suggestions.

Point 2. Overall, the papers first part (Fig 1) where the authors describe a connection between certain ECM and M0 macrophages seems weak as it relies too much on immune deconvolution analysis without even a basic marker-based sanity check on non-deconvoluted data to show whether M0 and indicated ECM markers co-occur in the studied or other omental HGSC cohort. The latter part with effects of decellularized models to macrophages, and these macrophages' effects to T cells is solid work and would be better off without the currently shaky basis that Fig. 1 forms.

Response. We have addressed these points with some additional analysis discussed below. Regarding the use of CIBERSORTx and xCell; these methods have both been validated comprehensively for deconvolution of whole tissue samples in their original papers and follow-up studies⁵⁻⁷. In Figure 1, the data presented is from a previous study² from where we have cell counts (IHC), proteomics, and transcriptomics on each tissue. Our analysis of IHC cell count correlations against CIBERSORTx and xCell in

this manuscript provides further confirmation that the *in silico* estimates vs actual cell counts match quite well, including macrophages that are the focus of the manuscript. The main message from Figure 1 is that M0 TAMs in HGSOC associate with a certain ECM composition, and this observation is tested in the figures that follow. We have answered specific comments below to hopefully provide further clarity about the IHC immune cell count correlations and have performed the further analyses requested.

Point 3. Description of the QC on deconvolution methods is insufficient and conclusions are not supported by the data:

“As anticipated, deconvolution of our bulk RNA-sequencing (RNAseq) dataset using analytical tools CIBERSORTx and xCell (which were designed to analyze RNAseq data) accurately estimated the abundance of CD8+ (R = 0.57 and 0.60, respectively), CD68+ (R = 0.36 and 0.46, respectively), CD3+, CD4+, and CD45RO+ immune cell counts.”

Response. We have compiled a table of the computed immune cell signatures which were summed to correlate against the IHC immune cell counts (Supplemental Table 1). In this process, we have found that three T cell signatures (Tregs, Th1 and Th2) were omitted from the previous sums which were correlated against CD3+ and CD4+ IHC immune cell counts. This has been corrected, and Supplemental Figure 1 Panel D and E have been updated to reflect this. The new analysis shows that xCell signatures significantly correlate with five out of six IHC immune cell counts (CD8, CD68, CD3, CD4 and CD45R, not FOXP3) and CIBERSORTx significantly correlates with CD8, CD68, and FOXP3 immune cell counts but only weakly correlates with CD3 and CD4, and does not correlate with CD45RO. Below is the new plots, and the changes to supplemental methods text.

Supplementary Figure 1. Validation of computational algorithms predicting transcriptomic immune landscape in HGSOC. D-E) Scatter plots of D) CIBERSORTx and CIBERSORT or E) CIBERSORTx and xCell correlations with previous immunohistochemistry staining cell counts using six immune markers present in CANBUILD dataset, in human HGSOC tissues. Regression analysis of summed abundances of all possible predicted immune cell types with each marker was completed to generate the plotted r values. Points are colored based on the significance of their r values from each technique, where $p < 0.05$ is significant. (Spearman's correlation, $N = 32$).

We have added in more detail in the supplementary methods and main text to clarify: Lines 33-56 - “**Merging cell subtype estimates for correlation to IHC marker cell counts.** We evaluated the performance of three deconvolution methods, CIBERSORT, CIBERSORTx, and xCell, by Spearman correlation between the cell type proportions computed by the different deconvolution methods and known compositions from IHC. From the default cell type estimations by CIBERSORT, CIBERSORTx and xCell, we identified all cell types that express each of the six immune cell markers for which cell count data was available (CD3, CD4, CD8, CD45RO, FOXP3, CD68) (Supplemental Table 1). CD3 cell scores are calculated as the sum of CD8 T cells, CD4 naïve T cells, $\gamma\delta$ T cells, CD4 memory T cells and follicular helper T cells (CIBERSORT and CIBERSORTx only) abundances; CD4 cell scores the sum of CD4 naïve T cells, T regs, CD4 memory cells, follicular helper T cells (CIBERSORT and CIBERSORTx only), $\gamma\delta$ T cells, in addition to CD4 Tem and CD4 Tcm for xCell only; CD8 cell scores as just CD8 T cells for CIBERSORT and CIBERSORTx, in addition to CD8 Tcm, CD8 Tem and CD8 naïve T cells for xCell. CD68 cell scores are calculated as the sum of monocytes, macrophages M0 (CIBERSORT and CIBERSORTx only) or macrophages (xCell only), M1 and M2, and dendritic cells (no immature DC); CD45RO just CD4 memory T cells and FOXP3 just Tregs. Spearman’s correlation analysis between the marker cell count data and the marker cell estimation scores was then performed (Supplemental Figure 1D and E). A limitation of our evaluation of the bulk deconvolution methods is that immune cell type proportions were assessed using IHC for only six markers: CD3⁺, CD4⁺, CD8⁺, CD68⁺, CD45RO⁺ and FOXP3⁺. As 22 immune cell types are computed by CIBERSORT and CIBERSORTx, and 34 immune cell types are computed by xCell, computed immune cell types were combined to assess their correlation against the IHC cell counts (e.g. xCell CD8⁺ naïve T cells, CD8⁺ Tcm and CD8⁺ Tem computed values were combined to correlate against CD8⁺ IHC immune cell counts). The gold standard to evaluate bulk deconvolution would be to compare against cell type proportions measured using single cell RNA seq or high resolution multiparameter flow cytometry, which has been performed comprehensively in the literature⁵⁻⁷.

Point 4. Only the CD8 predictions are relatively robust for all methods whereas no method predicts CD4⁺ correctly, and even CD68⁺ that should be relevant for any macrophage correlation analysis is relatively weak, although significant. The statement that also CD3⁺, CD4⁺, and CD45RO⁺ immune cell counts were accurately estimated is not supported by the data. Suppl.Fig 1D and 1E are not referred and miss description in main text/methods to explain how the comparison was made.

Response. We have corrected the text to accurately reflect Suppl. Fig 1D and 1E and provided more description for the comparison in Supplemental Methods (as above).

Main text:

Lines 96-104 - “As anticipated, deconvolution of our bulk RNA-sequencing (RNAseq) dataset using analytical tools CIBERSORTx and xCell (which were designed to analyze RNAseq data) accurately estimated the abundance of CD8⁺ ($p < 0.001$ and $p = 0.0001$, respectively), and CD68⁺ ($p < 0.05$ and $p < 0.01$, respectively) cells, confirming that expression of macrophage markers correlates with the immune cell counts by IHC (Supplemental Figure 1D-E). We cannot use IHC immune cell counts to verify the presence of all the immune subtypes, as we compare a restricted panel of six IHC markers against the 22 or 34 immune cell subtypes detected by CIBERSORTx and

xCell, however the computed subtypes are well-validated in the literature (Supplemental Methods, Supplemental Table 1).”

Point 5. Please more clearly state why “the composition of the tumor ECM associates most strongly with a M0 macrophage subtype”. This is not evident from Figs 1B-F.

Response. We have rephrased to clarify that “we found that the composition of the tumor ECM **which correlates with extent of disease** associates most strongly with a M0 macrophage subtype which is poorly characterized in HGSOc”. We have also rephrased the below text to make clear the analysis which shows this in Supplemental Figure 2:

Lines 117-122 - “**Level of disease and tumor ECM correlated most strongly** with the macrophage signatures described by CIBERSORTx ‘M0 macrophage’ and ‘Macrophage’ xCell signature, both of which are derived from the Immune Response In Silico (IRIS) gene expression dataset for macrophages after 7 days of differentiation from monocytes^{5,6,8} (Supplemental Figure 2A-B).”

Point 6. Please also directly assess expression of M0 markers in relation to ECM composition from transcriptomic (and if possible, also proteomic data), similar to seen in Fig 1D&E, especially as neither deconvolution method predicts the proportion of CD68+ cells particularly well.

Response. The correlation for CD68+ cells is significant between actual vs estimate counts using both CIBERSORTx and xCell (Supp Fig 1D). However we have conducted the analysis proposed in point 6, where we can see that COL11A1, MXRA5, SFRP2, FN1 and VCAN (the members of the ECM signature identified from Figure 1) at both the transcriptomic and proteomic level correlate strongly with several M0 macrophage markers (Response to Reviews Figure 1A-B). When we explore the number of significant associations between M0 markers and matrisome molecules, COL11A1 and VCAN are identified as the two matrisome molecules which significantly associate with the most M0 markers (Response to Reviews Figure 1C-D). SFRP2, MXRA5 and FN1 rank third, fourth and seventh, respectively (Response to Reviews Figure 1C-D).

Point 7. The M0 macrophage signature should be assessed in GEO available HGSC scRNA-seq or spatial transcriptomics data; do the M0 macrophages overlap with M2s or M1s?

Response. We used the Zhang *et al* scRNASeq dataset of metastatic ovarian cancer samples⁹ (GEO: GSE165897) to examine the expression of the M0, M1 and M2 macrophage signatures obtained from the CIBERSORTx LM22 gene-lists. The Zhang *et al.* study comprised of ovarian metastasis samples, including omental metastasis samples, obtained from treatment naïve patients and post-NACT from interval debulking surgery. All cell type annotation was retained from Zhang *et al.* and confirmed by Feature Plots and UMAPs illustrating the expression of cell type markers (Response to Reviews Figure 2A-B). We focused the analysis on the omental metastasis samples (although we observed very similar results also when using all samples of the Zhang *et al* study). Macrophage signature scores were obtained using Seurat’s AddModuleScore function. We have confirmed that the M0, M1 and M2 signatures are all expressed in the omental tumor-associated macrophage population (Response to Reviewers Figure 2C). We also observed that there is a modest but not clearcut compartmentalisation in their expression which supports a continuum shift of

the cells between these states. We are also observing a more wide-spread expression of the M0 signature compared to M1 and M2 signatures, which possibly indicates that the M0 gene list represents a more basal transcriptional program of macrophages, that captures tumor-associated macrophages.

Point 8. The term “poor prognostic M0 TAM signature” is misleading. The poor prognosis association is shown for ECM sub-signature (that associates with M0 in deconvoluted data) but not for M0 TAMs.

Response. We have corrected the sentence to read:

Line 169 - “poor prognostic **ECM signature associated with M0 TAMs** (Figure 1)”

Point 9. Higher percentage of PAX8+ cells and stroma in a tissue sample does not equal to disease progression; please rephrase: “ There was a significant difference in the disease scores between ECG1-2 when compared against ECG3, 4, or 5, indicating the change in ECM composition that occurs as disease progresses.” It appears that ECG1 & 2 simply define normal adipose tissue (and are later referred to as “adjacent tissue”); please refer to it as such or explain in the manuscript why this is not the case.

Response. We agree and have altered the text below accordingly:

Lines 238- “There was a significant difference in the disease scores between ECG1-2 when compared against ECG3, 4, or 5, indicating the change in ECM composition that occurs with **extent of disease. Tissue samples in ECG1-2 were mostly adipose tissue showing only a low/no level of tumor and stroma as observed from their disease score (Supplemental Figure 9D).**”

(Response to point 9 continued on page 10)

Response to Reviews Figure 1. Expression of M0 markers in relation to ECM composition from transcriptomic and proteomic data. A-B) Heatmap depicting Spearman correlation values between ECM and CIBERSORTx M0 macrophage transcriptomic signature markers for **A)** Transcriptomic ECM and **B)** Proteomic ECM. Spearman's r values are plotted according to the color scale, with positive correlations in red and negative in blue. Names of M0 macrophage-associated ECM molecules are in bold. Ordered by unsupervised clustering. **C)** Scatterplot depicting the number of CIBERSORTx M0 marker genes significantly correlated with matrisome molecules at protein and gene level. Points are colored based on the matrisome molecule as depicted in the key. Ranks are shown with dashed lines based on the color as depicted in the key. **D)** M0 macrophage associated matrisome molecules ranked in order of maximum number of significant positive associations.

Response to Reviews Figure 2. Expression of macrophage signatures in scRNASeq of HGSOc omental metastasis. The Zhang et al scRNASeq dataset of metastatic ovarian cancer samples (GSE165897) was analysed for the expression of macrophage signatures in omental metastasis samples (n = 13). A) UMAP illustrating cell subtypes of the immune compartment on omental metastasis samples from primary and/or interval debulking surgery. B) Features dot plot illustrating the relative expression of characteristic cell subtype genes across the immune cell clusters. C) UMAP illustrating expression of the macrophage markers CD68, MRC1 and CSF1R confirming confinement to the macrophage compartment. D) UMAP plot illustrating module scores of CIBERSORT's LM22 signatures for M0, M1 and M2 macrophages. B-D) colour-scales map colours to expression levels.

Response to point 9 continued. Regarding the low disease omental samples (ECG1-2), they were composed almost entirely of adipose tissue (Supplemental Figure 8D) and had ECM compositions comparable to normal tissues³ and low immune cell abundances (Figure 3E-F). However, since they are from cancer patients we do not want to describe them as normal. As such, tumor has been changed to 'high disease'

and adjacent has been changed to ‘low disease’ in the text and figures throughout. Similarly, MAM^{Tumor} has been replaced with MAM^{HD} (high disease extracellular matrix-associated macrophages) and MAM^{Adjacent} has been replaced with MAM^{LD} (low disease extracellular matrix-associated macrophages) throughout the text and figures.

Point 10. Figure 3 does not support the association between M0 macrophages and the ECM molecules of interest. In Fig 3E and 3F, all sample classes have macrophages as the most prominent class, and macrophage total count or proportion does not seem to be reflected in the strength of “M0 associated macrophage signature” proteins.

Response. We are not directly testing whether M0 macrophages associate with the ECM signature in this figure, we come on to show this in figure 4. In this figure we make two main points. Firstly tissues with high disease (ECG3-5) have higher immune cell infiltration vs tissues with low disease (ECG1-2). Secondly, high disease tissues with similar disease scores (ECG3-5) have different patterns of ECM composition, and ECG5 has the highest M0-ECM signature based on 4 molecules of the signature, and also the highest infiltrate of CD68+ cells.

Point 11. Supplemental Fig 19 and Fig 5 are too complicated for the reader with the color codes for WGCNA modules. Please name the clusters by functional annotations instead to help the reader.

Response. Figure 5 has been updated to give numbers to each Clusters instead of colors to improve clarity. It is not possible to name the clusters by functional annotations as the WGCNA analysis works by identifying clusters of highly correlated genes, but these can be across several functional pathways as illustrated in Figure 5D. Supplemental Fig 19 (now Supplemental Fig 20) figure legend has been edited to make clear that the colored WGCNA modules in this analysis are taken from published literature and applied to this dataset, as such it is not possible to re-name these clusters: Supplemental figure legend 21 “A) Heatmap showing correlation between colored module eigengenes (ME) taken from Xue et al., 2014”

Point 12. It is unclear from which data the gsva scores in Fig 5F are calculated; please add description.

Response. GSVA scores in Figure 5F were calculated from the MAM RNASeq dataset using the gene lists indicated next to the scatterplots and R package gsva. Gene lists were selected/ categorized based on their function/ literature. i.e., ECM collagen receptor signature combines various integrins and other collagen binding receptor genes; ECM collagen degradation signature combines various collagen degrading molecule genes; classic activation signature combines previously known M1-like macrophage activation genes

Point 13. The claim of tissue remodelling capabilities in M0 is supported by the data shown; modify statement to make claims of only tissue remodelling-related gene expression as the data supports that.

Response. The top DE genes associated with HD MAMs included immunoregulatory soluble factors like chemokine CXCL5, a chemoattractant of T cells, which was confirmed to be significantly upregulated as a secreted factor at the protein level using a multiplex assay. In addition, weighted gene correlation analysis identified that HD MAMs had upregulated programs enriched for toll-like receptor signaling. The data

supports MAM^{HD}s have an immunoregulatory phenotype, and may also have tissue remodelling capabilities (the latter we have not followed up on in this manuscript). We have clarified the text in Lines 336- “We selected some of the top DE genes including transmembrane receptors, immunoregulatory soluble factors, ECM-collagens, ECM-regulators, transcription factors, scavenger receptors like CD36, C-X-C motif chemokine ligand 5 (CXCL5), arginase (ARG1) and a number of matrix metalloproteinases (MMPs) (Figure 4I). The patterns of upregulated genes indicated HD MAMs may have immunoregulatory and ECM remodeling activities.” And from line 351 - “Gene programs such as clusters 8, 9, and 10 were significantly upregulated in HD MAMs, while gene programs contained within clusters 1, 2, and 12 and other WGCNA programs were downregulated (Figure 5C). HD MAMs upregulated programs (purple, tan and brown) were distinctly enriched for integrin receptors (blood coagulation pathway), toll-like receptor signaling,

Point 14. Fig 6 sub-panels and legend seem to be scrambled.

Response. Fig 6 sub-panels and legend have been corrected.

Reviewer #3 (expertise in ECM and decellularised tissue models):

Point 1. Overall, this manuscript by E. H. Puttock et al., describing a role for extracellular matrix proteins in the induction of immune suppression in human ovarian cancer, is novel, rigorous, and substantially advances our understanding of the role of the ECM in tumor cell immune surveillance. The core of the study is the prospective collection of fresh, patient derived ovarian cancer tissue specimens (HGSOC) and their analysis by gene expression and ECM proteomics. Using a multitude of RNA sequencing deconvolution and pathway analysis tools, associations between immune cell populations and ECM composition are elucidated. Further, how these associations change between adjacent normal, non-cancer pathologies and cancer are also delineated. Based on these omic-level studies, specific ECM compositions that correlated with tumor associated macrophages (TAMs) were identified. To demonstrate causality, fresh patient derived tissues were decellularized, and the resulting ECM scaffold used as a substratum to culture monocytes. In vitro, tumor derived matrices induced a macrophage phenotype similar to the TAM phenotype observed in patient samples. The authors refer to these ECM or matrix-educated macrophages as MAMs, and the determination that macrophages can be educated by human relevant, ovarian cancer ECM matrices is the major advance of this study. In an ex-vivo T cell activation assay (apparently antigen independent T cell activation, or at least in the absence of a model antigen), tumor educated MAMs induced T cell proliferation, activation and exhaustion markers more robustly than monocytes educated by decellularized adjacent normal tissues.

The paper has several strengths including the large number of patient derived tissues analyzed, and the inclusion of adjacent normal tissues and ovarian tissues with non-cancer pathologies. Another strength is the thoroughness by which the authors confirmed decellularization of the matrix scaffolds and demonstrate the integrity of these scaffolds after decellularization. Finally, a strength is the breadth of analyses employed, a requisite to undertake analysis of the tumor microenvironment in clinically relevant tissues.

The manuscript also has some significant weaknesses, all of which, in this reviewer's opinion, can be remedied by revision of the current document.

Response. Thank you for the positive comments on the work and the critical analysis of the manuscript. Below we have addressed the points raised.

Point 2. The primary weakness is the lack of clarity with respect to experimental design, making data interpretation difficult. The results, methods, and figure legend sections provide insufficient amount of information to interpret the data.

Response. We have added more detail into the appropriate sections throughout the main text and supplemental methods to improve the clarity of the experimental work undertaken. Changes in the text are highlighted in red text.

Point 3. Another weakness is the sparse discussion section, which mostly reiterates the results section.

- Key observations are not sufficiently discussed in light of the literature. Why might expression of their “core” ECM molecules FN1, COL11A1, VCAN, MXRA5, SFRP2, COL1A1, CTSB, and CS, be upregulated in ovarian cancer, and influence macrophage phenotypes?
- How might these molecules or the pathways they elicit be targeted, as the authors suggest?
- What are the limitations of their study? Due to low solubility, many ECM proteins are underrepresented in proteomic databases, and this is particularly true for the fibrillar collagens. A stepwise extraction with CHAPS and high salt, guanidine hydrochloride and chemical digestion with hydroxylamine hydrochloride (HA) has been shown to be optimal for ECM proteomics (see work of Kirk Hansen, for example <https://doi.org/10.1016/j.mcpro.2021.100079>). It appears from the methods section that this optimized approach was not used. Given the importance of the ECM proteomics data to the overall interpretation of the current study, additional information on sample preparation and inherent limitations to the ECM proteomics data need to be included in the methods and discussion sections, respectively.

Response. We have added more discussion to address these points now present within the discussion section. Thank you for recommending the paper by McCabe *et al*, this is a very useful manuscript to use when planning future ECM proteomics experiments and is now referenced in the discussion section.

Point 4. Referencing past studies, especially related to methods, are sparse. Pioneering work by others in the field of decellularized matrices are not referenced. Is the decellularization protocol used by the authors not based or inspired by any prior studies?

Response. We have added more references within the context of the work, and in particular we have highlighted the decellularization work by Laronda *et al*, Biomaterials, 2015, that inspired the model, and now mentioned in the discussion:

Line 426 – “...we developed a decellularized model of ovarian cancer omental metastatic tissue. Inspired from previous work decellularizing ovarian tissues in mice¹⁰, we optimized the methodology to decellularize human omental tissue whilst maintaining the native ECM composition and spatial arrangement of fibers.”

Point 5. The rationale and details of the phagocytosis assay are sparse, and again references to others performing these assays are lacking.

Response. We have added more detail regarding the phagocytosis assay as shown below and included references by Barkal *et al*, Nature, 2019, and Kelley *et al* EMBO Rep, 2021

Line 372 - we first performed a phagocytosis assay; **phagocytosis being an intrinsic function of macrophages^{11,12}. Briefly, MAMs were mixed with CTY⁺ K562 cells and flow cytometry was used to analyze the level of phagocytosis. HD MAMs had..**”

Line 542 - *Phagocytosis assay.* K562 cells **grown for 5 days and collected** from a T75 flask and then stained with CellTrace yellow (CTY) in PBS, 1:10,000 for 20 minutes at 37°C then 20mL RPMI was added and incubated 5 minutes at 37°C. CTY⁺ K562 cells were centrifuged and resuspended in RPMI at 5×10^5 cells/mL and plated onto an ultra-low attachment plate at 25,000 cells/well and incubated with anti-CD47 antibody at 40µg/mL for 1 hour at 37°C. In the meantime, **MAMs** were collected from decellularized tissues and seeded into a fresh ultra-low attachment plate at 10,000 cells/well. K562 cells/ well were washed in RPMI, resuspended in 100µL RPMI and combined with **MAMs** for 2 hours at 37°C. Cells were then washed and stained with a flow cytometry antibody cocktail and analysed by flow cytometry. Staining panel used: CD206 (FITC, 1:100), CD209 (APC, 1:100), HLA-DR (AF700, 1:100), CD36 (BV421, 1:100), CD38 (BV605, 1:150), CD86 (BV650, 1:150), CD11b (AF594, 1:200), CD14 (PE-Cy5, 1:150), CD204 (PE-Cy7, 1:100) and Zombie NIR (1:1000) (Biolegend); CTY was captured on the PE channel.

Line 838 - **Figure 6. HD MAMs have a reduced phagocytic response and alter T cell activation in the presence of CD3 and CD28 stimulation. MAMs were cultured for 14 days and K562 cells were cultured for 5 days prior to use in phagocytosis assay. Cell types were mixed for phagocytosis assay. A)** Flow cytometry analysis of mean fluorescence intensity (MFI) of **HD (MAM^{HD}) and LD MAMs (MAM^{LD}).**

Point 5. The methods for scanning electron microscopy and matrix fiber characterizations are missing.

Response. We have added the analysis method for quantifying ECM fiber diameter and alignment from SEM images to the main text methods and shown below:

Line 531 - *Quantifying ECM fiber diameter and alignment.* Quantification of fiber diameter and alignment was performed on SEM micrographs using ImageJ (NIH). Three fields of view were chosen at random in cellularized and matched decellularized tissue micrographs (N=15). Fiber diameter and fiber orientation angles were recoded (minimum 30 fibers quantified per field of view). The alignment index (AI) for each field of view was calculated as follows: where N is the total number of fibers quantified per field of view, θ is the fiber angle (rad), θ_{avg} is the average orientation angle among the collagen fibers of the said field of view. When collagen fibers are randomly aligned AI will equal 0, whereas aligned fibers will have AI = 1.

Point 6. T cell activation assay also is not referenced. What T cell population is isolated with this kit, CD3+, CD4+ or CD8+ T cells?

Response. Pan-T cell negative selection was performed by magnetic cell sorting (EasySep™ Human T Cell Isolation Kit from STEMCELL). CD3+ T cells were used in T cell activation assays.

We have added the following to the methods:

Lines 554– “Pan-T cell negative selection was performed by magnetic cell sorting (EasySep™ Human T Cell Isolation Kit from STEMCELL) from frozen PBMCs of autologous samples according to the manufacturer’s instructions. CD3⁺ T cells were then resuspended at 2 x 10⁶ cells/ml in PBS and stained with Cell Trace Violet (1:2000, Invitrogen) for 20 minutes at 37°C. Cells were then washed twice in complete medium (RPMI supplemented with 10% FBS and 1% Penicillin-Streptomycin), seeded in a 96-well plate at 2 x 10⁵ cells per well either alone or in presence of 20 x 10⁴ high or low disease MAMs and were activated with 1 µg/ml anti-CD3 and 5 µg/ml anti-CD28. At day five FACs analysis was performed to evaluate CD3⁺ T cell activation and proliferation.”

Point 7. Specific comments related to data presentation are as follows:

Figure 1

- Are Figures 1B & C combining proteomics ECM composition with immune gene signatures? Or are these analyses ECM gene expression to immune gene expression associations?

Response. This has been clarified in the text and figure to show that Figures 1B & C are combining proteomics ECM composition with immune gene signatures: Lines 129-132 - “we integrated sample matched CIBERSORTx and xCell immune signatures against gene (Supplemental Figure 3) and protein (Figure 1B-E) expression values for ECM molecules found in the matrisome database¹³, using Spearman correlative analysis”

- Is the primary goal to show correlations between ECM and immune signatures, or to show that these correlations change with disease state? It seems like the data presented in this figure are ‘richer’ than described.

Response. We were interested to determine if the type of immune infiltrate correlated with the type of ECM composition. We then looked at whether immune cell types consistently correlate with tumor ECM and increased level of disease (e.g. ‘M0 macrophages’ from CIBERSORTx and ‘Macrophages’ from xCell) and then to identify which ECM molecules consistently correlate with this immune cell type. In this case we found FN1, VCAN, MXRA5, COL11A1, and SFRP2 significantly correlate with CIBERSORTx ‘M0 macrophages’ and xCell ‘Macrophages’ at both the transcriptomic and proteomic level.

As pointed out, the data is richer than presented in the sense that there are other correlations that we are interested to investigate in the future.

- In Fig 1B & C, can the data be color coded by case/stage of disease? Rotating these figures to horizontal might help highlight the main point, which appears to be differential ECM proteins signatures between immune cell types? Are the authors suggesting that each immune cell subtype has its own ECM microenvironment? Or that different disease states have a unique ECM milieu, which associate with changes to the immune milieu? Please clarify.

Response. In Fig 1B & C, each color in the heatmap represents a correlation analysis across 32 samples with a range of disease stages and extent of disease, so it is not possible to color code the data. The main point has been clarified in the text, which is that there are changes in the composition of the ECM correlated with extent of disease and this associates with changes to the immune milieu:

Lines 162- “Taken together, we found that the composition of the tumor ECM **which correlates with extent of disease** associates most strongly with a M0 macrophage subtype which is poorly characterized in HGSOC”

- Fig 1E - are any of the immune cell separations observed within these two distinct clusters statistically different, or are the meaningful differences observed between the two main clusters?

Response. The scatterplot in Fig 1E depicts significant immune-matrisome correlations that occur at both the gene and protein level. There are significant positive correlations ($> \sim 0.4$ Spearman r value) and significant negative correlations ($< \sim -0.4$ Spearman r value) on this plot. These appear as distinct groups because insignificant correlations which would appear between the two groups ($< \sim 0.4$ and $> \sim -0.4$) have not been plotted.

- Fig 1G- based on proteomic analysis, 5 ECM proteins (FN1, COL11A1, VCAN, MXRA5, SFRP2) are used to delineate tumor high and low expression groups. Step 2 of the “cancer immunity cycle” is increased in high expressers. Do the authors know if all 5 ECM proteins are contributing to this difference or is one ECM protein dominating?

Response. We think it is the combination rather than one dominating protein. We have clarified this point in the text as follows:

Lines 146 – “TIP analysis also showed an increase in step 2, cancer antigen presentation, **which was not recapitulated in TIP analyses of the individual ECM proteins, suggesting that all five ECM proteins are contributing to this difference, so does not seem to be one particular protein but rather the pattern (Figure 1G, Supplemental Figure 5).**” Supplemental Figure 5 is shown here for convenience on page 19.

Figure 2

- Three of the 5 ECM molecules of focus change between Figure 1 (FN1, COL11A1, VCAN, MXRA5, SFRP2) and Figure 2D (COL1A1, FN1, CTSS, VCAN and CS) without a clear explanation for why.

Response. In Figure 2 we are focused on characterizing the integrity of the decellularized model and have focused the IHC characterization on the classes of matrisome molecule that make up the tissue (collagens, glycoproteins, proteoglycans, secreted factors, and major post-translational molecules) with the goal of testing whether the decellularization method is resulting in loss of these molecules.

- Figure 2F & G - methods for quantifying fiber diameter and alignment are missing.

Response. We have added this information to the main text methods as shown below

Line 531 - **Quantifying ECM fiber diameter and alignment.** Quantification of fiber diameter and alignment was performed on SEM micrographs using ImageJ (NIH). Three fields of view were chosen at random in cellularized and matched decellularized tissue micrographs (N=15). Fiber diameter and fiber orientation angles were recorded (minimum 30 fibers quantified per field of view). The alignment index (AI) for each

field of view was calculated as follows: where N is the total number of fibers quantified per field of view, θ is the fiber angle (rad), θ_{avg} is the average orientation angle among the collagen fibers of the said field of view. When collagen fibers are randomly aligned AI will equal 0, whereas aligned fibers will have $AI = 1$.

- Figure 2F- presumably the red cells are dying? Please clarify in the figure legend. It is unclear to this reviewer whether these data are simply artifacts of the cell culture system or of use to the interpretation of the in vivo biology they are trying to dissect.

Response. We have added to the description of the figure which now reads:
Line 784- **J**) Representative **live (green)/ dead (red)** immunofluorescence (IF) images from decellularized tissue model cultures using monocytes/ macrophages at day 1 and 7

Figure 3

- It is unclear what is meant by disease score (disease profiling)-are these clinical parameters of disease such as tumor size, stage or a molecular signature? What does synergy mean in the figure title?

Response. The disease score is a histopathological analysis of the tissue which is the sum of tumor cell area (by PAX8+) and stromal area. We first reported it in². It does not have any association with tumor size or staging. It is a way to characterise tissues prior to further analysis. We have added more detail in the main text to clarify the point:

Line 230 – “**The disease score is a digital histopathology method we described previously², that measures extent of disease present within a sample based on the area of tumor (PAX8 positivity) and stroma (hematoxylin staining of areas where there is stromal remodeling or existing stroma). Previously we found disease score to be associated with an increase in immune infiltration, tissue stiffness, and ECM remodeling.**”

- Are the data in Fig 3B defining ECG1-5, or have these categories been previously defined? The figure legend in Fig 3B reads as if these clusters were known previously, but the data appear to be used to define these 5 groups. Please clarify.

Response. Yes the data in 3B are defining ECG clusters. We have clarified this in main text to read:

Line 792 -”**...B**) Hierarchical unsupervised clustering (ward.D2 method) separated **the tissue samples into five groups, based on the samples ECM protein expression, that we have termed as ECM composition groups (ECG) 1-5.**”

- Is the M0 macrophage-associated ECM signature based on gene expression? The Y axis reads as if only FN1, COL11A1, VCAN, and MXRA5 protein levels are being quantified, and not an M0 gene signature. Please clarify.

Response. This is a proteomic ECM signature associated with M0 macrophages. The Y axis and title of Figure 3, Panel D has been updated to clarify this point.

- How were immune cells/tissue in Fig 3E assessed, flow? IHC? State in figure legend or results text.

Response. We have clarified this in the text to now read:

Line 799 – “... **E) Immune cells were detected by IHC and analyzed by QuPath; mean number of immune cells/ number of tissues between ECG1-5 (N = 38).”**”

Figure 4

- Figure 4B- in the legend, please state that the clustering is of monocytes after 14 days in culture on tumor and adjacent decellularized donor tissue and provide numbers of cases per condition. Were any replicates done to confirm low intra-assay variability?
- Figure 4F-is this flow data? The legend says histograms of receptor expression.
- Figure 4G- presumably the low/low and high/high populations are either rare or did not change between tumor and adjacent decellularized matrices?
- Figure 4H has two panels, and the distinction is not discussed in the text or legend. It is unclear what “donor” refers to in this figure.
- Figure 4I-are these top 30 genes found to be enriched in MAMS, meaning expressed preferentially in monocytes cultured onto tumor decellularized stroma? Please clarify.

Response. We have clarified the following points for figure 4 within the legend which now reads:

Line 803 - **Figure 4. Tumor ECM alters the macrophage transcriptome. A)** Schematic of macrophage decellularized tissue culture. **Monocytes/ macrophages from four separate blood donors were cultured for 14 days on high disease (MAM^{HD}) (N = 4) or low disease (MAM^{LD}) (N = 4) decellularized tissues (total N = 31, 4 blood donors, x4 high disease samples (minus G198_D3 (Supplemental Figure 19B)), x4 low disease samples). Each sample represents six (wells) pooled culture samples from a 96-well plate, each well containing 200,000 cells. B)** Unsupervised cluster dendrogram using DE genes **from HD or LD MAMs. C-E)** Bar plots of CIBERSORTx analysis using DE genes between **HD and LD MAMs for C) M0 macrophage, D) M1 macrophage and E) M2 macrophage signatures. F) Flow cytometry histograms** of transmembrane receptors expressed on **HD and LD ECM cultured macrophages. Representative contour plot of HD and LD MAMs. G) Bar plots of flow cytometry expression patterns of CD163 and CD209 between HD and LD MAMs, shown as percentage from the CD45⁺ CD14⁺ macrophage population. For example, CD163^{low} and CD209^{high} populations were selected from the bottom-right gating (M1-like) and CD163^{high} and CD209^{low} populations were selected from the top-left gating (M2-like) from Figure 4F contour flow cytometry plots. Mann-Whitney U test, p = 0.004 and Unpaired T test, p = 0.0016, respectively (N = 3). H) Heatmap of row z-scores of log2TPM gene expression for top 30 (right panel) up- and downregulated genes; total (left panel) up- and down-regulated genes (adj. p < 0.05, logFC > |1|, protein coding). Samples split by ECM type (LD (blue) or HD (brown)) and donor 1-4 i.e., blood donors. I) Bar plot of selected DE genes (from the list of top 30) between HD versus LD cultured macrophages. J) LEGENDPLEX™ assay of secreted CXCL5. Each dot represents the mean value of sample duplicates for each blood donor per tissue (N = 32). Mann-Whitney U test, p < 0.0001. K) LEGENDPLEX™ assay of secreted CXCL1 and CCL2 expression levels.**

Supplemental Figure 5. Boxplot of predicted tumor immune phenotype stage scores by TIP. HGSOc samples separated by high (red) and low (blue) protein expression levels of **A) FN1, B) VCAN, C) COL11A1, D) MXRA5, E) SFRP2.** Scores are based upon expression of signature genes and represent activity levels. $N = 32$. Two-way ANOVA significance between each group is presented as $*p < 0.05$.

Figure 5

- I don't find the color coding of data to be highly informative-I think Figure 5B and C do not add much value. It is unclear why some gene clusters (colors) are

described more fully than others. For example, why is the gene set defined as turquoise not discussed?

Response. Colour coding is used to highlight gene groups which contain hundreds of genes and multiple biological processes. The data is presented here is standard for this type of analysis, and examples of other papers presenting WGNA data can be found here¹⁴⁻¹⁷, and requires extraction to be more informative which we have done as the figure progresses. The processes are extracted from some of these color groups selected based on their significance values. 5A and B provide an overview of the WGCNA data, and provide a visual representation of the relationship between genes, and these types of plots are used routinely, we do think they are useful to keep in the figure to orientate the reader to the data being presented.

- Figure 5E seems largely unrelated to the WGCNA data analyses, but important none-the less. Please provide more details on how this select group of MAM signature genes were collated.

Response. We have added additional details to explain how we selected this group. The text now reads:

Line 360 – “We built a **HD MAM** gene signature based on DE genes and TAM-like markers **from the transcriptomic data set** (Figure 5E)...”

- Figure 5F- it is unclear what is being correlated in these graphs-gene expressing to gene expression scores obtained fromwhat samples?

Response. GSVA scores in Figure 5 F were calculated from the MAM RNASeq dataset using the gene lists indicated next to the scatterplots and R package gsva.

Figure 6

- The overall design of this experiment is vague. Are macrophages engulfing CTY labeled tumor cells in an ex vivo assay? Are live or dead tumor cells added? Are these data consistent with data shown in figure 1G, where antigen presentation is reported to be increased?

Response. We have added more detail regarding the phagocytosis assay as shown below and included references by Barkal et al, Nature, 2019, and Kelley et al EMBO Rep, 2021

Line 372 - we first performed a phagocytosis assay; **phagocytosis being an intrinsic function of macrophages^{11,12}. Briefly, MAMs were mixed with CTY⁺ K562 cells and flow cytometry was used to analyze the level of phagocytosis.**

Line 547 - **Live** K562 cells/ well were washed in RPMI, resuspended in 100µL RPMI and combined with **MAMs** for 2 hours at 37°C.

Line 838 - **Figure 6. HD MAMs have a reduced phagocytic response and alter T cell activation in the presence of CD3 and CD28 stimulation. MAMs were cultured for 14 days and K562 cells were cultured for 5 days prior to use in phagocytosis assay. Cell types were mixed for phagocytosis assay.**

Regarding the data shown in figure 1G, where antigen presentation is reported to be increased. The TIP analysis in Figure 1G explores the effect of the M0 macrophage-

associated matrix signature on the tumor immunity cycle, other immune cell types could be contributing to this increased antigen presentation. Interestingly, correlative analysis between xCell computed immune cell types and the HGSOc matrisome identifies B cells as significantly positively correlated at both the transcriptomic and proteomic level with COL11A1, MXRA5, VCAN, and SFRP2, but not FN1 (Figure 1C, Supplementary Figure 3B), which would be interesting to explore in the future but is not a focus of this manuscript.

- Figure 6D, E & F-please label control conditions on X-axis. Please clarify in results text or figure legend that data shown in D, E and F were obtained by flow cytometry.

Response. For Figure 6D, E & F, control conditions have been labelled “T cells only” on the X-axis and this has been clarified in the figure legend text. The figure legend text has also been clarified to state that the data shown in D, E and F were obtained by flow cytometry:

Lines 834 - “**D)** Normalized MFI of LAG3, PD1 and TIM3 expression on T cells as assessed **using flow cytometry** after 5 days **culture alone** or co-culture with high or low disease MAMs. ****p < 0.01, *p < 0.05**, one-way ANOVA followed by Dunnett’s multiple comparisons test. **E)** Percentage of proliferating T cells as assessed by CTV dilution **using flow cytometry** after 5 days **culture alone** or co-culture with high or low disease MAMs ***p < 0.05**, one-way ANOVA followed by Dunnett’s multiple comparisons test. **F)** Normalized MFI of LAG3, PD-1 and TIM3 expression on T cells and percentage of proliferating T cells as assessed **using flow cytometry** after 5 days **culture alone** or co-culture with high or low disease MAMs conditioned media.”

Reviewer #4 (expertise in ECM biology and ECM in cancer):

Point 1. This manuscript examines the matrisome in high grade serous ovarian cancer (HGSOc) omental metastases and also uses a decellularized/re-cellularized tissues to probe the ability of the matrix to “educated” tumor associated macrophages (TAMs). Overall, the work is thorough, performed rigorously, and presents convincing results to largely support the authors conclusions. However, I have a few comments that warrant some attention.

Response. *We thank the reviewer for their positive appraisal of the work and have addressed the comments below.*

Point 2. I have several questions regarding the fibers in the SEMs. The identities are unclear as in some of the panels they appear to be fibrillar collagen and in others more like basal lamina (Fig 2e). Yet the fibrillar orientation analysis appears to be the same. I recommend some immunostaining to identify these and clarify the analysis. Similarly, is there sufficient data to perform quantitative analysis of fiber? Lastly, as there have been several reports using optical microscopy of collagen alignment in the fallopian tubes, ovarian cortex and omentum, is the alignment data similar to those results?

Response. To explore the alignment further we have stained a set of four tissues with Masson’s trichrome, which reveals the collagen fibers well, and have used TWOMBLI to analyse the fiber alignment after decellularization. We have not detected any differences between cellularized and decellularized tissues across the parameters

TWOMBLI measures. We have added this work to the manuscript through new text and figure panels:

Lines 191-201 - “We next tested whether decellularization affected the ECM architecture by scanning electron microscopy (SEM) (Figure 2E, Supplemental Figure 11B), and compared ECM fiber diameter and alignment between matched cellularized and decellularized tissues, using ImageJ digital software analysis. We also performed Masson’s trichrome stain to detect collagen ECM architecture and used the TWOMBLI plugin to quantify differences in alignment (Figure 2H-I, Supplemental Figure 12). Taken together, we observed no change between cellularized and decellularized tissues for both low and high disease samples which displayed highly aligned fibers, consistent with the literature demonstrating aligned collagen fibers in HGSOC ovarian tissue using second harmonic generation imaging (Figure 2F-I) (PMID: 20222963, 34199725).”

Point 3. I would like to see more on the criterion for identifying regions that are tumor and tumor adjacent (as used in several figures). While a mainly cellular criterion was used, it has been shown in the FT and primary ovary that collagen can be altered in regions of low cellularity, where the fiber morphology is highly distinct from normal tissues or distant normal regions in diseased tissues. Thus, more rigorously establishing this classification is important as it was used in a few contexts.

Response. We agree that ‘adjacent’ is not the correct description of what these samples are, which is omental samples with a low level of disease as detected by histopathological analysis. The tissue samples of ECG1-2 were composed almost entirely of adipose tissue (Supplemental Figure 8D) and had ECM compositions comparable to normal tissues³ and low immune cell abundances (Figure 3E-F). To make this clearer in the text and figures we have changed where we have used tumor and adjacent to ‘high disease’ and ‘low disease’ respectively. Similarly, MAM^{Tumor} has been replaced with MAM^{HD} (high disease extracellular matrix-associated macrophages) and MAM^{Adjacent} has been replaced with MAM^{LD} (low disease extracellular matrix-associated macrophages) throughout the text and figures.

Point 4. The large results showing differences in Col11 in low and high disease in the hazard analysis (Supp. Fig 7b) are interesting. There are several reports in the literature regarding Col 11 expression and remodeling, and it would be ideal to put the current findings in that context.

Response. We have added further discussion on the proteins identified including COL11A1, which reads as follows:

Line 413 – “COL11A1 is associated with poor prognosis in several cancer types, and may have potential as a prognostic biomarker in pancreatic cancer⁴⁷. Its function within the tumor microenvironment is not well investigated, but may be connected with activation of fibroblasts into cancer-associated fibroblast (CAF) phenotype⁴⁸, and some reports indicate COL11A1 expression to be a CAF-specific feature⁴⁹. We were surprised to see COL11A1 upregulated in ECG 1, a group of tissues with low disease and at least from histopathological analysis appear normal and similar to the other low disease group ECG2. This may indicate this group of tissues are already undergoing extracellular matrix remodeling predisposing them for tumor colonization, or may be due to a separate parameter such as patient age. These concepts were recently investigated using bulk and single cell transcriptomics datasets of squamous cell carcinomas⁵⁰ where ECM changes were found to be predictive of premalignant

progression, of which COL11A1 was one of several markers identified. Comparing the age of patients, ECG1 tissues had a trend to be from younger individuals.”

- 1 Gyorffy, B., Lanczky, A. & Szallasi, Z. Implementing an online tool for genome-wide validation of survival-associated biomarkers in ovarian-cancer using microarray data from 1287 patients. *Endocr Relat Cancer* **19**, 197-208 (2012). <https://doi.org:10.1530/ERC-11-0329>
- 2 Pearce, O. M. T. *et al.* Deconstruction of a Metastatic Tumor Microenvironment Reveals a Common Matrix Response in Human Cancers. *Cancer Discov* **8**, 304-319 (2018). <https://doi.org:10.1158/2159-8290.CD-17-0284>
- 3 Naba, A. *et al.* Characterization of the Extracellular Matrix of Normal and Diseased Tissues Using Proteomics. *Journal of proteome research* **16**, 3083-3091 (2017). <https://doi.org:10.1021/acs.jproteome.7b00191>
- 4 Parker, A. L. *et al.* Extracellular matrix profiles determine risk and prognosis of the squamous cell carcinoma subtype of non-small cell lung carcinoma. *Genome Med* **14**, 126 (2022). <https://doi.org:10.1186/s13073-022-01127-6>
- 5 Aran, D., Hu, Z. & Butte, A. J. xCell: digitally portraying the tissue cellular heterogeneity landscape. *Genome Biol* **18**, 220 (2017). <https://doi.org:10.1186/s13059-017-1349-1>
- 6 Newman, A. M. *et al.* Determining cell type abundance and expression from bulk tissues with digital cytometry. *Nat Biotechnol* **37**, 773-782 (2019). <https://doi.org:10.1038/s41587-019-0114-2>
- 7 Steen, C. B., Liu, C. L., Alizadeh, A. A. & Newman, A. M. Profiling Cell Type Abundance and Expression in Bulk Tissues with CIBERSORTx. *Methods Mol Biol* **2117**, 135-157 (2020). https://doi.org:10.1007/978-1-0716-0301-7_7
- 8 Abbas, A. R. *et al.* Immune response in silico (IRIS): immune-specific genes identified from a compendium of microarray expression data. *Genes Immun* **6**, 319-331 (2005). <https://doi.org:10.1038/sj.gene.6364173>
- 9 Zhang, K. *et al.* Longitudinal single-cell RNA-seq analysis reveals stress-promoted chemoresistance in metastatic ovarian cancer. *Sci Adv* **8**, eabm1831 (2022). <https://doi.org:10.1126/sciadv.abm1831>
- 10 Laronda, M. M. *et al.* Initiation of puberty in mice following decellularized ovary transplant. *Biomaterials* **50**, 20-29 (2015). <https://doi.org:10.1016/j.biomaterials.2015.01.051>
- 11 Barkal, A. A. *et al.* CD24 signalling through macrophage Siglec-10 is a target for cancer immunotherapy. *Nature* **572**, 392-396 (2019). <https://doi.org:10.1038/s41586-019-1456-0>
- 12 Kelley, S. M. & Ravichandran, K. S. Putting the brakes on phagocytosis: "don't-eat-me" signaling in physiology and disease. *EMBO Rep* **22**, e52564 (2021). <https://doi.org:10.15252/embr.202152564>
- 13 Naba, A. *et al.* The matrisome: in silico definition and in vivo characterization by proteomics of normal and tumor extracellular matrices. *Mol Cell Proteomics* **11**, M111 014647 (2012). <https://doi.org:10.1074/mcp.M111.014647>
- 14 Maniati, E. *et al.* Mouse Ovarian Cancer Models Recapitulate the Human Tumor Microenvironment and Patient Response to Treatment. *Cell Rep* **30**, 525-540.e527 (2020). <https://doi.org:10.1016/j.celrep.2019.12.034>
- 15 Feng, Y., Li, Y., Li, L., Wang, X. & Chen, Z. Identification of specific modules and significant genes associated with colon cancer by weighted gene co-expression network analysis. *Mol Med Rep* **20**, 693-700 (2019). <https://doi.org:10.3892/mmr.2019.10295>

- 16 Cui, W. *et al.* Discovery and characterization of long intergenic non-coding RNAs (lincRNA) module biomarkers in prostate cancer: an integrative analysis of RNA-Seq data. *BMC Genomics* **16 Suppl 7**, S3 (2015).
<https://doi.org/10.1186/1471-2164-16-S7-S3>
- 17 Seefelder, M. & Kochanek, S. A meta-analysis of transcriptomic profiles of Huntington's disease patients. *PLoS One* **16**, e0253037 (2021).
<https://doi.org/10.1371/journal.pone.0253037>

REVIEWERS' COMMENTS

Reviewer #1 (expert in ECM characterisation, ECM proteomics):

The authors have adequately address all of my suggestions and concerns.

Reviewer #2 (expert in ovarian cancer RNA-seq, single-cell transcriptomics):

The authors' responses and revisions are comprehensive and address previous review comments.

Minor: Sentence on line 429 has repetition (comparison... compared) and needs to be fixed.

Reviewer #3 (expert in 3D ECM culture models, ECM proteomics, cancer metastasis):

The core of this impressive study by E. H. Puttock et al., is the prospective collection of fresh, patient derived ovarian cancer tissue specimens (HGSOC), their analysis by gene expression and ECM proteomics, and use to generate decellularized matrices to assess ex-vivo macrophage and T cell phenotypes. This revised manuscript, describing a role for extracellular matrix proteins in the induction of immune suppression in human ovarian cancer, is novel, rigorous, and substantially advances our understanding of the role of the ECM in tumor cell immune surveillance. Most prior concerns, of which were dominantly focused on the lack of clarity with respect to experimental design, methodologic details, and lack of detail in the figure legends, have been adequately addressed. One relatively minor concern remains, and that is to provide, in the Discussion, a better integration to published TAM work. In fact, some of this information is already included in the Results section, and could be moved to the Discussion for improved clarity. Additional comments:

Lines 121-125: Are there any relationships between previously published immature myeloid and/or MDSC gene signatures and the CIBERSORTx MO macrophage and or MAM gene signatures described here?

Lines 310-313: The spectrum model of macrophage polarization is impressive and informative. Revisiting these data in the Discussion, similar to the suggestions above regarding published work on immature and MDSC populations (lines 121-125), would help put these newly defined MAMs into context.

Paragraph 217: The 5 ECM composition groups (ECG1-5) are intriguing. Are the primary differences between ECM protein abundance or are there also unique compositions? Some discussion dedicated to the implications of these 5 ECM profiles, and next steps with respect to their prognostic significance, would help round out the discussion. For example, can the ECG 1-5 compositions distinguish between good and poor prognosis similar to the overall ECM signature associated with M0 macrophages (Supplemental Figure 7B)?

Line 339, reference to Fig 4I. Please label the Y axis so it is evident that it is the ratio of MAM HD/MAM LD gene expression being assessed.

Line 372 and method section for phagocytosis assay: Phagocytosis assay (Figure 6A) is not well described. Please explain why K562 cells were selected, how they were labeled (CTY is not mentioned in the methods), and how the phagocytic events were captured. If this is a standard assay, referencing past work will be sufficient.

Line 462-464: is this unpublished data from the PI's lab? Probably best to omit, as robustness cannot be assessed.

Lines 422-425: A limitation to the study is lack of adjacent or true normal, and this should be acknowledged, as the potential explanation for why COL1A1 is upregulated in ECG1, i.e., due to the tissue being at risk for progression, is a hypothesis that requires normal tissue for validation. Minor typos, grammatical errors. For example, line 29, is missing "that" between population & correlated, and line 379, should this read CD209+ and CD209-? Please check for other grammatical errors.

Reviewer #4 (expert in ECM biology, ECM and ovarian cancer):

The authors have sufficiently addressed all my concerns and suggestions.

RESPONSE TO REVIEWERS' COMMENTS

Reviewer 3

Comment 1. The core of this impressive study by E. H. Puttock et al., is the prospective collection of fresh, patient derived ovarian cancer tissue specimens (HGSOC), their analysis by gene expression and ECM proteomics, and use to generate decellularized matrices to assess ex-vivo macrophage and T cell phenotypes. This revised manuscript, describing a role for extracellular matrix proteins in the induction of immune suppression in human ovarian cancer, is novel, rigorous, and substantially advances our understanding of the role of the ECM in tumor cell immune surveillance. Most prior concerns, of which were dominantly focused on the lack of clarity with respect to experimental design, methodologic details, and lack of detail in the figure legends, have been adequately addressed. One relatively minor concern remains, and that is to provide, in the Discussion, a better integration to published TAM work. In fact, some of this information is already included in the Results section, and could be moved to the Discussion for improved clarity.

Response. We have revised the manuscript to include further consideration of published TAM work in the discussion, as outlined below.

Comment 2. Lines 121-125: Are there any relationships between previously published immature myeloid and/or MDSC gene signatures and the CIBERSORTx M0 macrophage and or MAM gene signatures described here?

Response. Myeloid derived suppressor cells (MDSC) are a heterogenous population of immature myeloid cells that include monocytic (M-MDSC) and granulocytic polymorphonuclear (PMN-MDSC) subsets which both have strong immunosuppressive potential and constitute a major component of TME (PMID: 33445053). Macrophages differentiated from M-MDSCs have substantially higher expression of S100A9, NOS2, ARG1, SIGLEC10 and S100P than macrophages generated from monocytes of the same patients (PMID: 33378668). S100A9, NOS2, ARG1, SIGLEC10 and S100P do not feature in the CIBERSORTx M0 macrophage signature or xCell macrophage signature (Supplemental Table 4). In the HD MAMs, ARG1 is significantly up-regulated compared to LD MAMs (3.7 logFC), but SIGLEC10 and S100A9 are significantly down-regulated (-0.63 and -0.91 logFC, respectively), and NOS2 and S100P are not significantly differentially expressed. Taken together, these data suggest that there does not appear to be a clear relationship between immature myeloid cells and the MAM phenotypes generated in this work.

Comment 3. Lines 310-313: The spectrum model of macrophage polarization is impressive and informative. Revisiting these data in the Discussion, similar to the suggestions above regarding published work on immature and MDSC populations (lines 121-125), would help put these newly defined MAMs into context.

Response. We have added the following. Lines 458-465: Analysis using a spectrum model of macrophage polarization revealed that neither MAM phenotype most strongly associates with stimuli linked to M1 or M2 polarization, which was in line with the analysis on whole cancer tissues (Figure 1). These data suggest that HD and LD MAMs undergo transcriptional reprogramming, however, overall these changes are not consistent with a strict M1-M2-axis re-polarization. In line with these data, there is increasing evidence in the literature that whilst most TAMs are polarized towards an M2-like phenotype²³⁻²⁴, they also exist as a spectrum of subtypes influenced by their interactions within the TME^{16-17,39}.

Comment 4. Paragraph 217: The 5 ECM composition groups (ECG1-5) are intriguing. Are the primary differences between ECM protein abundance or are there also unique compositions? Some discussion dedicated to the implications of these 5 ECM profiles, and next steps with respect to their prognostic significance, would help round out the discussion. For example, can the ECG 1-5 compositions distinguish between good and poor prognosis similar to the overall ECM signature associated with M0 macrophages (Supplemental Figure 7B)?

Response. The following has been added to the discussion. Lines 414-433: Proteomic analysis on our library of ovarian metastatic tissues revealed five ECM composition groups (ECGs) which clustered based on the relative expression of ECM proteins, rather than the presence or absence of specific proteins. Therefore, it is the pattern of ECM that separates these groups. In the low disease tissues, we found two ECGs (ECG1-2), and we were surprised to see COL11A1 upregulated in ECG1, a group of tissues which at least from histopathological analysis appear normal and similar to the other low disease group ECG2. This may indicate this group of tissues are already undergoing ECM remodeling predisposing them for tumor colonization, or may be due to a separate parameter such as patient age. These concepts were recently investigated using bulk and single cell transcriptomics datasets of squamous cell carcinomas⁵² where ECM changes were found to be predictive of premalignant progression, of which COL11A1 was one of several markers identified. Comparing the age of patients, ECG1 tissues had a trend to be from younger individuals. Further investigations to explore early changes in tumour matrix or changes with

age would benefit from the analysis of true healthy control tissue from cancer free patients. High disease tissues were separated into three ECGs (ECG3-5) based on the pattern of ECM detected, which correlated with immune infiltrate, and in particular a high macrophage presence in tissues composed of high M0-ECM. These composition signatures could be reflected in the circulating fragments of ECM detectable in blood or urine and may have utility in cancer prognostics or diagnostics. There is significant interest in using ECM for prognostics and diagnostics, particularly within the collagen family of molecules (PMID: 33687786, PMID: 26406420, PMID: 24261855, PMID: 27465284, PMID: 31642523, PMID: 31875000).

Comment 5. Line 339, reference to Fig 4I. Please label the Y axis so it is evident that it is the ratio of MAM HD/MAM LD gene expression being assessed.

Response. Fig 4I: The Y axis has been labelled to be clear that the ratio of MAM HD/MAM LD gene expression is being assessed.

Comment 6. Line 372 and method section for phagocytosis assay: Phagocytosis assay (Figure 6A) is not well described. Please explain why K562 cells were selected, how they were labeled (CTY is not mentioned in the methods), and how the phagocytic events were captured. If this is a standard assay, referencing past work will be sufficient.

Response. We have made the following revisions to the text and methods. Line 370: "Briefly, MAMs were mixed with CellTrace Yellow (CTY) stained K562 cells and flow cytometry was used to measure the number of phagocytic events (i.e. CTY+ MAMs)."

Method section for phagocytosis assay: "K562 cells, an immortalized myelogenous leukemia cell line which are lysed rapidly by Fc receptor-positive leukocytes (i.e. sensitive for killing assays) and do not phagocytose or mediate antibody-dependent phagocytosis (PMID: 789258, PMID: 14629626, PMID: 15728242), were grown for 5 days and collected from a T75 flask and then stained with CellTrace Yellow (CTY) in PBS, 1:10,000 for 20 minutes at 37°C then 20mL RPMI was added and incubated 5 minutes at 37°C."

Comment 7. Line 462-464: is this unpublished data from the PI's lab? Probably best to omit, as robustness cannot be assessed.

Response. This is an unpublished observation and has been removed as suggested.

Comment 8. Lines 422-425: A limitation to the study is lack of adjacent or true normal, and this should be acknowledged, as the potential explanation for why COL1A1 is upregulated in ECG1, i.e., due to the tissue being at risk for progression, is a hypothesis that requires normal tissue for validation.

Response. The following edit has been made in the discussion on Line 425-427: "Further investigations to explore early changes in tumour matrix or changes with age would benefit from the analysis of true healthy control tissue from cancer free patients."

Comment 9. Minor typos, grammatical errors. For example, line 29, is missing "that" between population & correlated, and line 379, should this read CD209+ and CD209-? Please check for other grammatical errors.

Response. We have made the following edits.

Line 29: We have rewritten to read "...a tumor-associated macrophage (TAM) population associated with poor prognosis..."

Line 377: We have corrected this to read CD209⁺ or CD206⁺ macrophages as this analysis explores populations positive for either CD209⁺ or CD206⁺.

We have checked for other grammatical errors and corrected these throughout.